# LaSe-E2V: Towards Language-guided Semantic-aware Event-to-Video Reconstruction

**Kanghao Chen**[1]    **Hangyu Li**[1]    **Jiazhou Zhou**[1]    **Zeyu Wang**[2,1,3]    **Lin Wang**[1,2,3*]

[1]AI Thrust, [2]CMA Thrust, HKUST(GZ)    [3]Dept. of CSE, HKUST

kchen879@connect.hkust-gz.edu.cn, linwang@ust.hk

Project Page: https://vlislab22.github.io/LaSe-E2V/

## Abstract

Event cameras harness advantages such as low latency, high temporal resolution, and high dynamic range (HDR), compared to standard cameras. Due to the distinct imaging paradigm shift, a dominant line of research focuses on event-to-video (E2V) reconstruction to bridge event-based and standard computer vision. However, this task remains challenging due to its inherently ill-posed nature: event cameras only detect the edge and motion information locally. Consequently, the reconstructed videos are often plagued by artifacts and regional blur, primarily caused by the ambiguous semantics of event data. In this paper, we find language naturally conveys abundant semantic information, rendering it stunningly superior in ensuring semantic consistency for E2V reconstruction. Accordingly, we propose a novel framework, called LaSe-E2V, that can achieve semantic-aware high-quality E2V reconstruction from a language-guided perspective, buttressed by the text-conditional diffusion models. However, due to diffusion models' inherent diversity and randomness, it is hardly possible to directly apply them to achieve spatial and temporal consistency for E2V reconstruction. Thus, we first propose an Event-guided Spatiotemporal Attention (ESA) module to condition the event data to the denoising pipeline effectively. We then introduce an event-aware mask loss to ensure temporal coherence and a noise initialization strategy to enhance spatial consistency. Given the absence of event-text-video paired data, we aggregate existing E2V datasets and generate textual descriptions using the tagging models for training and evaluation. Extensive experiments on three datasets covering diverse challenging scenarios (*e.g.*, fast motion, low light) demonstrate the superiority of our method. *Demo videos for the results are attached to the project page.*

## 1 Introduction

Event cameras are bio-inspired sensors that detect per-pixel intensity changes, producing asynchronous event streams [5] with high dynamic range (HDR) and high temporal resolution. They particularly excel in capturing the edge information of moving objects, thus discarding the redundant visual information. Such a distinct imaging shift poses challenges for integration with the off-the-shelf vision algorithms designed for standard cameras. To bridge the event-based and standard computer vision [27, 31, 3, 29], a promising way is event-to-video (E2V) reconstruction.

Recently, deep learning has been applied to E2V reconstruction [49, 54, 64], and remarkable performance is achieved thanks to the availability of synthetic event-video datasets and the development of model architectures. Most existing research emphasizes on the network design [64, 13, 54, 34] or high-quality data synthesis [57, 34]. However, this task remains challenging due to its inherently ill-posed nature: event cameras capture edge and motion information locally but neglect semantic

---

*Corresponding author

38th Conference on Neural Information Processing Systems (NeurIPS 2024).

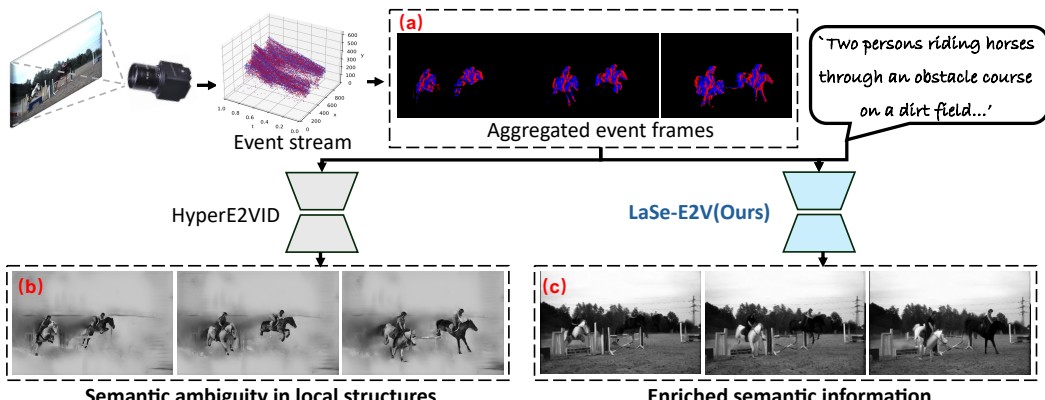

Figure 1: Comparison of the E2V pipeline between HyperE2VID [13] and our LaSe-E2V: The baseline method solely relies on event data, leading to ambiguity in local structures. In contrast, our approach integrates language descriptions to enrich the semantic information and ensure the video remains coherent with the event stream.

information in regions with no intensity changes (See Fig. 1 (a)). Consequently, the reconstructed videos, *e.g.*, HyperE2VID [13], are plagued by artifacts and regional blur, as shown in Fig. 1 (b).

To fill this gap, in this paper, we find language naturally conveys abundant semantic information, which is beneficial in enhancing the semantic consistency for the reconstructed video, as shown in Fig. 1 (c). Intuitively, we propose a novel language-guided semantic-aware E2V reconstruction framework, called **LaSe-E2V**, with the text-conditional diffusion model [52] as the backbone. While many efforts [62, 51, 23] apply diffusion models for images and video generation, adapting them to our problem is hardly possible. The reason is that the inherent randomness in diffusion sampling and the diversity objectives of generative models may result in temporal inconsistency between consecutive frames and spatial inconsistency between the event data and reconstructed videos.

Therefore, we first introduce an Event-guided Spatio-temporal Attention (ESA) module to enhance reconstructed video by introducing spatial and temporal event-driven attention layers, respectively (Sec. 3.2). This module not only achieves fine-grained spatial alignment between the events and video (See Fig. 7 (right)) but also maintains temporal smoothness and coherence by adhering to the temporal properties of event cameras. We then introduce an event-aware mask loss to maintain temporal coherence throughout the video by considering the spatial constraints of event data from adjacent frames (Sec. 3.3). Lastly, we propose a noise initialization strategy that utilizes accumulated event frames to provide layout guidance and reduce discrepancies between the training and inference stages of the denoising process (Sec. 3.4).

Given the absence of event-text-video paired data, we aggregate existing E2V datasets and employ tagging models to generate textual descriptions, thereby facilitating both training and evaluation processes. Extensive experiments on three widely used real-world datasets demonstrate the superiority of our method in enhancing the quality and visual effects of reconstructed videos, especially its super generalization ability when applied in challenging scenarios, e.g., fast motion, and low light (See Fig. 4 and Fig. 5). In summary, the contributions of our work are three-fold: (**I**) We explore E2V reconstruction from a language-guided perspective, utilizing the text-conditioned diffusion model to effectively address the semantic ambiguities inherent in event data. (**II**) We propose the event-guided spatio-temporal attention mechanism, an event-aware mask loss, and a noise initialization strategy to ensure the semantic consistency and spatio-temporal coherency of the reconstructed video. (**III**) We have rebuilt event-text-video paired datasets based on existing event datasets with textual descriptions generated from off-the-shelf models [73]. Extensive experiments on three datasets covering diverse scenarios (*e.g.*, fast motion, low light) demonstrate the effectiveness of our framework.

## 2 Related Works

**Event-to-Video (E2V) Reconstruction.** E2V reconstruction falls into two categories: model-based and learning-based methods. Model-based approaches [2, 44, 6, 53] exploit the correlation

between events and intensity frames through hand-crafted regularization techniques. However, its reconstruction result is comparatively inferior to the more recent learning-based methods [50, 49, 54, 34, 13]. For example, E2VID [50, 49] used an Unet-like network with skip connections and ConvLSTM units to reconstruct videos from long event streams. Following E2VID, SPADE-E2VID [7] extended E2VID to enhance temporal coherence by feeding previously reconstructed frames into a SPADE block. HyperE2VID [13] introduced hypernetworks to generate per-pixel adaptive filters guided by a context fusion module. GANs, such as conditional GAN [61] and cycle-consistency GAN [69], are used to address this issue, but often exhibit blurry images with artifacts in non-activated areas. In summary, previous learning-based methods can only achieve plausible reconstructed results for this ill-posed problem because event data solely captures motion information without semantic context information. In this work, *we explore the possibility of incorporating textual descriptions with semantic awareness for E2V reconstruction.*

**Text-to-Video Diffusion Models.** The success of Diffusion models [45, 52] in generating high-quality images from text prompts has advanced text-to-image (T2I) synthesis. Inspired by T2I synthesis, text-to-video (T2V) diffusion models, such as [4, 19, 20, 23, 30, 40, 55, 60, 66] adapt T2I synthesis to video. Subsequent developments [23, 4, 18] focused on refining the temporal information interaction, such as optimizing temporal convolution or self-attention modules for motion learning. Meanwhile, one T2V research line focuses on enhancing controllability by incorporating additional conditions, thus addressing the text prompts' ambiguity in motion, content, and spatial structure. For high-level video motion control, studies propose learning LoRA [24] layers for specific motion patterns [18, 75], or utilizing extracted trajectories [68], motion vectors [62], pose sequences [41] to guide the synthesis. For fine-grained spatial structure control, methods like Gen-1 [14], VideoComposer [62], and others [42, 70] leverage monocular depth, sketch, and image control models [42, 70] for flexible and controllable video generation [8, 18, 30, 72].

Formally, given a video sample $x_0$, the latent diffusion model (LDM) [52] first encodes it into a latent feature $z_0 = \mathbf{E}_I(x_0)$. A noisy input is obtained based on the forward diffusion process by introducing Gaussian noise to the latent representation: $z_t = \sqrt{\bar{\alpha}_t} z_0 + \sqrt{1 - \bar{\alpha}_t} \epsilon$, where $t = 1, ..., T$ and $T$ denotes the maximum timestep. $\bar{\alpha}_t = \prod_{i=1}^{t}(1 - \beta_i)$ is the coefficient that controls the noise strength. The diffusion model is trained by predicting the noise $\epsilon$ through a mean squared error:

$$l_\epsilon = \|\epsilon - \epsilon_\theta(z_t, \mathbf{c}, t)\|^2, \tag{1}$$

where $\theta$ denotes the parameters of the U-Net of the diffusion model. $\mathbf{c}$ denotes the addition conditions (text, image, depth, *et al.*) to control the diffusion process. In the inference stage, the generated sample $\hat{x}_0$ can be obtained from the denoised latent $\hat{z}_0$ using a pre-trained decoder $\hat{x}_0 = \mathbf{D}_I(\hat{z}_0)$.

These methods achieve fine-grained controllability based on the well-designed condition $\mathbf{c}$, focusing on flexible user instructions and diverse outcomes. In this study, we employ a basic T2V diffusion model with the 3D-UNet [60] architecture as the foundation for language-guided E2V reconstruction. In particular, *our framework prioritizes video fidelity by aligning motion details from event data and semantic insights from text, thus outperforming previous models, especially in extreme scenarios like fast motion and low light.*

## 3 The Proposed LaSe-E2V Framework

**Event Representation.** An event stream $\mathcal{E} = \{e_{t_k}\}_{k=1}^{N_e}$ consists of $N_e$ events. Each event $e_{t_k} \in \mathcal{E}$ is represented as a tuple $(x_k, y_k, t_k, p_k)$, which denotes the pixel coordinates, timestamp, and polarity. To make the event stream compatible with the pre-trained diffusion model, we convert $\mathcal{E}$ into a grid-like event voxel grid $V \in \mathbb{R}^{B \times H \times W}$ with $B$ time bins using temporal bilinear interpolation [77].

### 3.1 Overall Pipeline

The goal of our LaSe-E2V is to reconstruct a video $\hat{\mathbf{x}} = \{\hat{x}^1, \hat{x}^2, ..., \hat{x}^N\} \in \mathbb{R}^{N \times C \times H \times W}$ from the set of event segments $\{\mathcal{E}^i\}_{i=1}^{N}$, where $\mathcal{E}^i$ is $i$-th event segment corresponding to the frames. The reconstructed video is expected to retain the motion and edge content from the event data and compensate for the semantic information from the language description. The difficulty of our task lies not only in achieving high visual quality that aligns with the text descriptions but also in maintaining content consistency with the event data. We address the difficulty by integrating the diffusion model

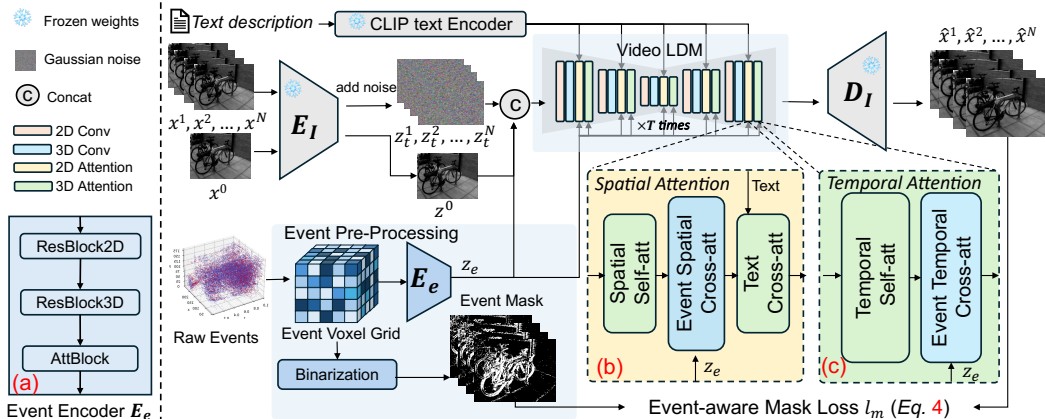

Figure 2: An overview of our proposed LaSe-E2V framework.

with events and text descriptions effectively. As shown in Fig. 2, LaSe-E2V consists of four major components: an image encoder $\mathbf{E}_I$, an event encoder $\mathbf{E}_e$, a video LDM, and a decoder $\mathbf{D}_I$.

**Image Encoder $\mathbf{E}_I$.** Given a sequence of frames $\mathbf{x} = \{x^1, x^2, ..., x^N\}$, we extract the latent representation $\mathbf{z} = \{z^1, z^2, ..., z^N\}$ with the pre-trained image encoder of LDM [52].

**Event Encoder $\mathbf{E}_e$.** To adapt the event voxel grids to the latent space of $\mathbf{E}_I$, we first project it into a latent representation with an event encoder $\mathbf{E}_e(\mathbf{V})$. As illustrated in Fig. 2 (a), we initially apply lightweight spatial 2D and temporal 3D convolution blocks to the set of input voxel $\mathbf{V} = \{V^i\}_{i=1}^N$ with $N$ segments. These blocks are designed to extract local spatial and temporal feature information, which is then processed through an attention block (*AttBlock*) for global temporal modeling. Subsequently, we concatenate the event latent representation with the noise latent representation along the channel dimension to serve as a rough condition for the model. Finally, we construct the input latent representation as $\mathbf{z_t} = \{[z_t^i, z_e^i]\}_{i=1}^N \in \mathbb{R}^{N \times (C' + C_e) \times H' \times W'}$, where $z_t^i$ and $z_e^i$ denote the noise latent and event feature corresponding to the $i$ frame with channel dimensions $C'$ and $C_e$.

**Video LDM.** We first extend the capabilities of the text-conditional image-based LDM by incorporating temporal layers in the U-Net, following the approach of video diffusion models [14, 23, 55, 62]. As depicted in Fig. 2, our framework presents a video LDM with multiple blocks consisting of 2D spatial convolution, 3D temporal convolution, 2D spatial attention, and 3D temporal attention layers. A pre-trained CLIP [48] text encoder is employed to extract text descriptions conditioning on the attention layers to provide semantic information for the reconstructed video. To facilitate fine-grained control of event data, we propose an Event-guided Spatio-temporal Attention (ESA) module (see Fig. 2 (b) and (c)) following $\mathbf{E}_e$ to convert the event voxel grids to the latent space. The ESA module enhances the control by introducing a specific spatial and temporal cross-attention in the U-Net. The technical details will be discussed in Sec. 3.2.

**Decoder $\mathbf{D}_I$.** Finally, we apply the pre-trained decoder $\mathbf{D}_I$ of [52] to convert the estimated latent representation $\hat{\mathbf{z}}$ to video $\hat{\mathbf{x}}$ in the image space.

**Event-aware Mask Loss $l_m$.** To optimize LaSe-E2V, we propose a novel event-aware mask loss $l_m$, in addition to the $\epsilon$-prediction loss (Eq. 1). The details of $l_m$ will be described in Sec. 3.3.

In addition, different from previous LDM models [10, 51], we introduce a noise initialization strategy to alleviate the train-test gap in the inference stage, which will be described in Sec. 3.4.

### 3.2 Event-guided Spatio-temporal Attention (ESA)

Relying solely on the 3D U-Net and feature concatenation is insufficient to reconstruct a content-consistent video with fine-grained control from event data, as demonstrated experimentally in Fig. 7 (right). To address this, we propose using the event latent representation to condition spatial and temporal attention, ensuring more reliable spatial alignment and temporal consistency (See Tab. 4). Our ESA module is specially designed to enhance the spatio-temporal consistency between events

and video in contrast to the original SD [52], which serves as a baseline attention mechanism for integrating conditional input. Our approach differs in attention design, which introduces two distinct attention mechanisms respective to the spatial domain and temporal domain.

**Event Spatial Cross-Attention.** As illustrated in Fig. 2 (b), the 2D spatial attention layer in the LDM U-Net integrates a self-attention mechanism that processes each frame individually and a cross-attention mechanism connecting frames with text embedding, which follows Stable Diffusion [52]. Intuitively, Event data naturally deliver abundant edge information in the spatial space, which is expected to directly constrain the structural information of the reconstructed video. To incorporate event information in the spatial attention module, we concatenate the features from the event latent $z_e^i$ to the U-Net intermediate features $z$ to formulate a event-based spatial attention:

$$z_{out} = Softmax \left( \frac{\mathbf{Q}\mathbf{K}_{e-s}^T}{\sqrt{d}} \right) \mathbf{V}_{e-s},\qquad(2)$$

where $\mathbf{Q} = \mathbf{W}^Q z^i$, $\mathbf{K}_{e-s} = \mathbf{W}^K [z^i, z_e^i]_s$, $\mathbf{V}_{e-s} = \mathbf{W}^V [z^i, z_e^i]_s$. $[*]_s$ represents the concatenation operation on the *spatial* dimension. This modification ensures that each spatial position in all frames accesses comprehensive information from the corresponding event data, enabling detailed structural feature control within the spatial attention layers.

**Event Temporal Cross-Attention.** In addition to the spatial information, event data inherently represents the difference between the adjacent frames, which is a strong constraint. As shown in Fig. 2 (c), to effectively leverage the temporal constraint of the event data, we facilitate event features into temporal attention. Specifically, given the intermediate features $z$, we first reshape the height and width dimensions into the batch dimension, forming a new hidden state $\bar{\mathbf{z}} \in \mathbb{R}^{(H \times W) \times N \times C}$. Then the event latent feature $z_e^i$ is employed to interact with the latent feature $\bar{z}$ in the temporal dimension through the temporal attention layer:

$$z_{out} = Softmax \left( \frac{\mathbf{Q}\mathbf{K}_{e-t}^T}{\sqrt{d}} \right) \mathbf{V}_{e-t},\qquad(3)$$

where $\mathbf{Q} = \mathbf{W}^Q \bar{z}^i$, $\mathbf{K}_{e-t} = \mathbf{W}^K [\bar{z}^i, z_e^i]_t$, $\mathbf{V}_{e-t} = \mathbf{W}^V [\bar{z}^i, z_e^i]_t$. $[*]_t$ represents the concatenation operation on the *temporal* dimension.

**Previous Frame Conditioning.** To ensure consistency for the whole video reconstruction, we use an autoregressive way to reconstruct the video, by conditioning the model on the last frame $\hat{x}^N$ of the previously estimated video clip $\hat{\mathbf{x}}$. We concatenate the previous frame with the latent representation $\hat{\mathbf{z}}$ on the temporal dimension as a condition to obtain $\bar{z} = [z_0^0, \mathbf{z_t}] \in \mathbb{R}^{(N+1) \times C' \times H' \times W'}$. Due to the absence of the previous frame in the first video clip, we randomly drop the previous frame condition by applying Gaussian noise to the frame latent representation. During inference, the first video clip is reconstructed directly from the events and text data without the previous frame. For subsequent video clips, the reconstruction process utilizes the last frame from the previous video clip as a condition to ensure temporal coherency and smoothness.

## 3.3 Event-aware Mask Loss

Considering the noise prediction loss can not capture the event constraint on temporal coherence, we introduce an event-aware mask loss to directly supervise the inter-frame difference:

$$l_m = \left\| (1 - \mathcal{M}) \cdot (\hat{z}_0^t - \hat{z}_0^{t-1}) \right\|_2^2,\qquad(4)$$

where $\mathcal{M} = I(\mathbf{V})$ indicates the event motion mask obtained by setting value one for event activated area and value zero for non-activated part; $\hat{\mathbf{z}}_0$ represents the model's estimated clean video latent, which can be obtained by:

$$\hat{z}_0 = \frac{z_t - \sqrt{1 - \bar{\alpha}_t}\epsilon_\theta(z_t, t, \mathbf{c})}{\sqrt{\bar{\alpha}_t}}.\qquad(5)$$

Finally, we combine the noise prediction loss and the event-aware mask loss with the scaling factor $\lambda$.

$$l = l_\epsilon + \lambda \cdot l_m.\qquad(6)$$

Table 1: Quantitative comparison with state-of-the-art methods on both synthetic and real-world benchmarks. The best and second best results of each metric are highlighted in **red** and blue, respectively. To align the metric of SSIM, we re-evaluate the previous methods based on their pre-trained models to obtain SSIM$^*$.

| Datasets | Metrics | E2VID [50] | FireNet [54] | E2VID+ [57] | FireNet+ [57] | SPADE-E2VID [7] | SSL-E2VID [46] | ET-Net [64] | HyperE2VID [13] | LaSe-E2V (Ours) |
|---|---|---|---|---|---|---|---|---|---|---|
| ECD | MSE↓ | 0.212 | 0.131 | 0.070 | 0.063 | 0.091 | 0.046 | 0.047 | 0.033 | **0.023** |
| | SSIM↑ | 0.424 | 0.502 | 0.560 | 0.555 | 0.517 | 0.364 | 0.617 | 0.655 | - |
| | SSIM$^*$↑ | 0.450 | 0.459 | 0.503 | 0.452 | 0.461 | 0.415 | 0.552 | 0.576 | **0.629** |
| | LPIPS↓ | 0.350 | 0.320 | 0.236 | 0.290 | 0.337 | 0.425 | 0.224 | 0.212 | **0.194** |
| MVSEC | MSE↓ | 0.337 | 0.292 | 0.132 | 0.218 | 0.138 | 0.062 | 0.107 | 0.076 | **0.055** |
| | SSIM↑ | 0.206 | 0.261 | 0.345 | 0.297 | 0.342 | 0.345 | 0.380 | 0.419 | - |
| | SSIM$^*$↑ | 0.241 | 0.198 | 0.262 | 0.212 | 0.266 | 0.264 | 0.288 | 0.315 | **0.342** |
| | LPIPS↓ | 0.705 | 0.700 | 0.514 | 0.570 | 0.589 | 0.593 | 0.489 | 0.476 | **0.461** |
| HQF | MSE↓ | 0.127 | 0.094 | 0.036 | 0.040 | 0.077 | 0.126 | 0.032 | **0.031** | 0.034 |
| | SSIM↑ | 0.540 | 0.533 | 0.643 | 0.614 | 0.521 | 0.295 | 0.658 | 0.658 | - |
| | SSIM$^*$↑ | 0.462 | 0.422 | 0.536 | 0.474 | 0.405 | 0.407 | 0.534 | 0.531 | **0.548** |
| | LPIPS↓ | 0.382 | 0.441 | **0.252** | 0.314 | 0.502 | 0.498 | 0.260 | 0.257 | 0.254 |

## 3.4 Event-aware Noise Initialization

During training, we construct the input latent by adding noise on the clean video latent. The noise schedule leaves some residual signal even at the terminal diffusion timestep $T$. As a result, the diffusion model has a domain gap to generalize the video during the inference time when we sample from random Gaussian noise without any real data signal. To solve this train-test discrepancy problem, during the testing, we obtain the base noise by adding noise on event-accumulated frames using the forward process of DDPM [21]. The noise latent for frame $i$ can be expressed as:

$$z_T^t = \sqrt{\alpha_T} z_e^t + \sqrt{1 - \alpha_T} \epsilon^i, \tag{7}$$

where $\alpha_T$ denotes the diffusion factor and $z_e^t = \mathbf{E}_I(I^{-1} + \sum_0^t e^i)$. The $I^{-1}$ is the last frame of the previous estimated video clip. Intuitively, accumulated event data provides structural information (*e.g.*, edges) in the scene, acting as an additional spatial constraint during the denoising process.

# 4 Experiments

## 4.1 Datasets and Implementation Details

**Dataset.** We train our pipeline using both synthetic and real-world datasets. For synthetic data, following prior arts [50, 34], we generate event and video sequences from the MS-COCO dataset [33] using the v2e [25] event simulator because it ensures stable and high-quality ground truth images. To enrich semantic information, we also utilize the real-world dataset BS-ERGB [58], which includes 1k training sequences. For all datasets, we employ the off-the-shelf tagging model RAM [73] to generate language descriptions. Recent work [65] has demonstrated the superiority of RAM compared to other prompting models due to its rich objects and concise description. We evaluate our model on Event Camera Dataset (ECD) [43], Multi Vehicle Stereo Event Camera (MVSEC) dataset [76] and High-Quality Frames (HQF) dataset [57]. RAM is also used to generate tags for these datasets.

**Implementation Details.** Based on Stable Diffusion 2.1-base [52], we use a text-guided video diffusion model [51] to initialize our model, pre-trained on large-scale video datasets [1]. For each training video clip, we sample 16 frames and the corresponding event streams, with an interval of $1 \leq v \leq 3$ frames. The input size is adapted to $256 \times 256$. Following previous methods [64, 57], the data augmentation strategies include Gaussian noise, random flipping, and random pause. The value $\lambda$ is set to 0.01 for all experiments. The model is trained with the proposed loss across all U-Net parameters, with a batch size of 3 and a learning rate of 5e-5 for 150k steps on 8 NVIDIA V100 GPUs. We also provide an analysis of the computation complexity. *See more details in Appendix 5.*

**Evaluation Metric.** The Mean Squared Error (MSE), Structural Similarity (SSIM [63]), and Perceptual Similarity (LPIPS [71]) are used to measure image quality. The SSIM metric raises ambiguity as it involves several hyperparameters that may differ across various codebases. For this reason, we reevaluated all compared methods using a unified metric, denoted as the SSIM$^*$ scores.

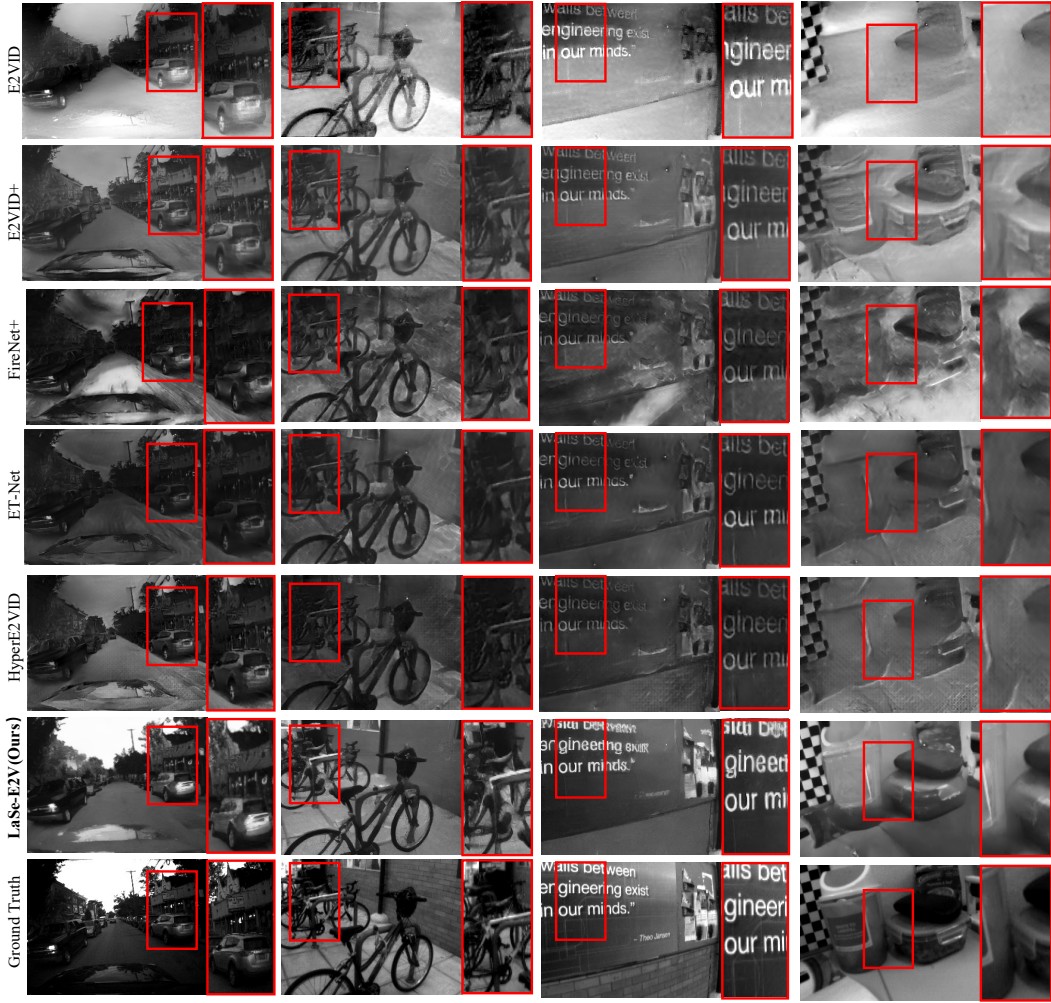

Figure 3: Qualitative comparisons on four sampled sequences from the test datasets. While the previous approaches suffer from low contrast, blur, and extensive artifacts, LaSe-E2V obtains clear edges with high contrast and preserves the semantic details of the objects

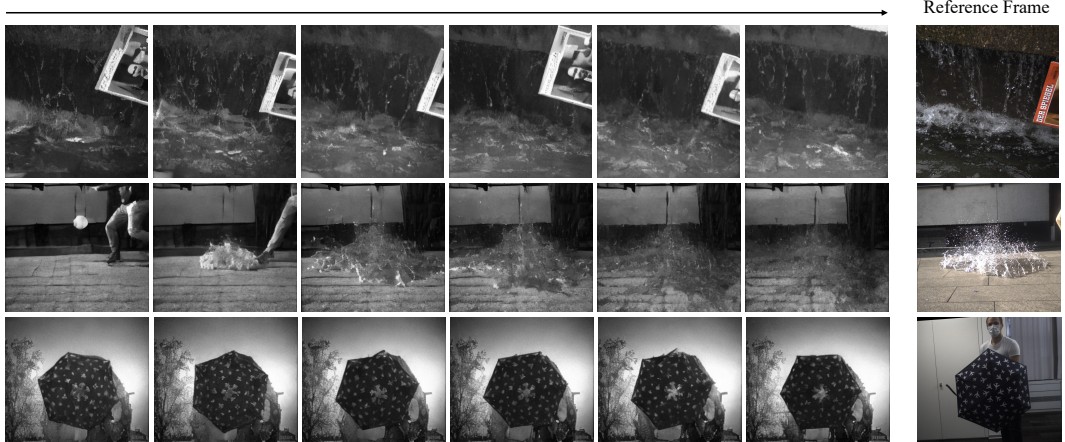

Figure 4: Qualitative results of fast-motion condition from HS-ERGB dataset [58].

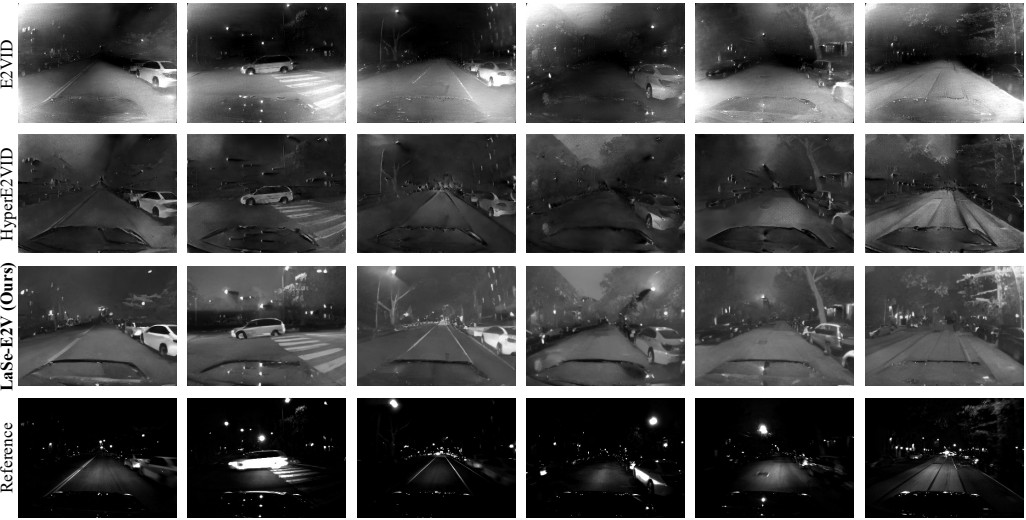

Figure 5: Qualitative results in low light condition from MVSEC dataset [76] (*outdoor_night2*). LaSe-E2V performs better to preserve the HDR characteristic of event cameras with higher contrast.

## 4.2 Comparison with State-of-the-Art Methods

**Quantitative results.** We compare LaSe-E2V with eight existing learning-based methods [50, 54, 57, 57, 7, 46, 64, 13]. To ensure fair comparison, we maintain identical experiment settings without any post-processing operations across all methods. As shown in Tab. 1, LaSE-E2V mostly outperforms previous methods. In particular, LaSE-E2V significantly surpasses HyperE2VID by 30% on MSE on the ECD dataset. LaSE-E2V also excels in SSIM and LPIPS mostly. This highlights the superiority of the structural and semantic reconstruction ability of LaSE-E2V.

**Qualitative Results.** Fig. 3 illustrates the qualitative results reconstructed by our LaSe-E2V and previous methods. As shown in the street reconstruction (Column 1), our method reconstructs more details, especially for the car and the building. For the bike in Column 2, our method achieves clearer edges and higher contrast, while the previous methods are inclined to exhibit foggy artifacts around the bikes. Although our method achieves superior performance, some artifacts persist for the reconstruction of the text region (*e.g.*$3^{rd}$ row). This issue arises because it depends on the prior of the pre-trained diffusion model (SD2 [52]), which faces challenges in text generation. The recently released SD3 [15] claims to show improved text generation capabilities, which could potentially address the problem. ***Please refer to the video demo and the appendix for additional qualitative results***, which demonstrates the superiority of our LaSe-E2V on temporal smoothness and consistency.

**Results with Fast Motion.** In Fig. 4, we show sampled reconstructed frames based on sequences of HS-ERGB [58] captured by Prophesee Gen4 ($1280 \times 720$) event camera with high resolution and relatively fast motion conditions. We can see that our method is capable of clearly recovering the details for high-speed movement and preserving temporal consistency.

**HDR Results.** We test our model on the video sequences in extreme conditions (*i.e.*low light and fast motion) to further demonstrate the advantages of event cameras and the effectiveness of our framework. As shown in Fig. 5, we sample sequences from MVSEC (*outdoor_night2*) with relatively low-light conditions in nighty streets. We can see that LaSe-E2V performs better in reconstructing the scene with higher contrast and more clear edge. Compared with HyperE2VID exhibiting foggy artifacts of the whole street, E2VID can reconstruct video without over-exposure or under-exposure.

**Quantitative Result on Temporal Consistency.** We further evaluate the results based on the temporal quality metrics from VBench [26]. As shown in Tab. 2, the numerical results demonstrate the effectiveness of our approach in maintaining temporal consistency. In particular, our method significantly outperforms others on the subject consistency and background consistency, while achieving comparable performance on motion smoothness.

Timestamp           Reference Frame

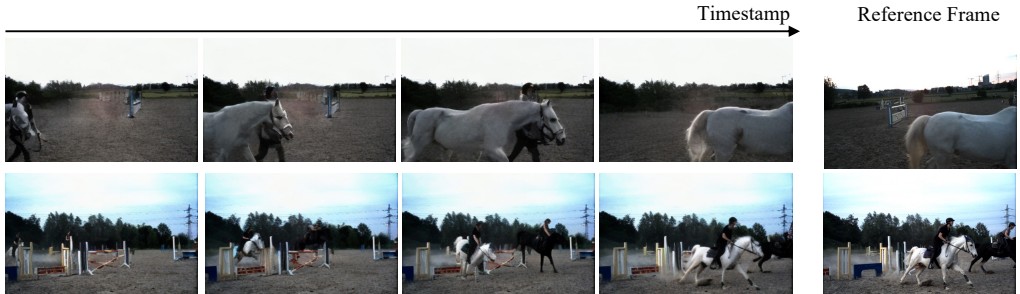

Figure 6: Qualitative results for color video reconstruction.

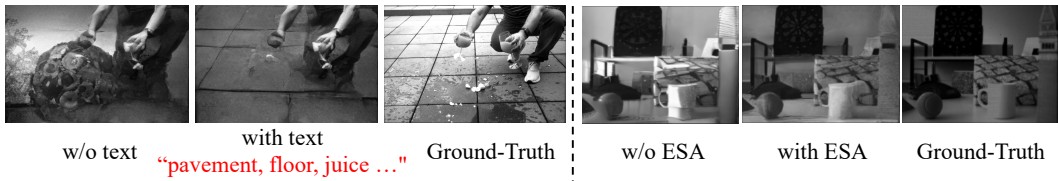

w/o text    with text "pavement, floor, juice …"    Ground-Truth  |  w/o ESA    with ESA    Ground-Truth

Figure 7: Qualitative comparison for the ablation study on text guidance and ESA module.

Table 2: Quantitative comparison on temporal consistency on ECD [43] based on VBench [26].

| Metrics | E2VID [50] | FireNet [54] | SPADE-E2VID [7] | SSL-E2VID [46] | ET-Net [64] | HyperE2VID [13] | LaSe-E2V (Ours) | GT (Empirical Max) |
|---|---|---|---|---|---|---|---|---|
| Subject Consistency↑ | 52.14% | 49.61% | 50.58% | 51.89% | 55.49% | 50.41% | **84.25%** | 88.29% |
| Background Consistency↑ | 85.26% | 82.78% | 82.61% | 84.86% | 86.85% | 83.50% | **93.39%** | 93.65% |
| Motion Smoothness↑ | 97.62% | 98.40% | **98.41%** | 95.97% | 97.72% | 97.59% | 98.11% | 98.67% |

## 4.3 Discussion

**Video Editing with Language.** In our framework, language serves as supplementary semantic information for E2V reconstruction. We investigate the impact of varying language guidance for video editing, as illustrated in Fig. 8. Utilizing a text description from the off-the-shelf tagging model [73], our method reconstructs a scene with a reasonable structure in low-light conditions based on descriptors like "*night, dark, ...*". Interestingly, when we manually alter the text description to "*bright, day light, ...*", the lighting condition in the scene shifts to daylight, revealing clearer details, especially in the sky area. This demonstrates our framework's ability to modify lighting conditions based on textual descriptions, by effectively modeling light conditions as semantic information. In this way, with a language-guided perspective, it offers the flexibility to manually adapt the reconstructed video according to user preferences.

"night, dark, city street…"    "bright, day light, city street …"    Reference Frame

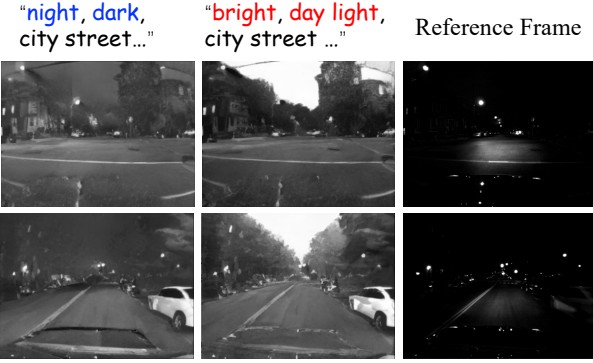

Figure 8: Qualitative results for video editing with text.

**Color Video Reconstruction.** Based on a pre-trained diffusion model [51], our model inherits the capability for colored video generation by training on real-world datasets. As illustrated in Fig. 6, our method successfully reconstructs color videos with clear details, although it occasionally misinterprets background semantics due to the absence of event data.

## 4.4 Ablation Study

We conducted ablation experiments on the LaSe-E2V framework to evaluate the effectiveness of language guidance, event-based attention, event-aware mask loss, and initialization strategy.

**Language Guidance.** To evaluate the effectiveness of language guidance, we conducted experiments without text conditions by setting the text input to null during the denoising stage. As shown in Tab. 3, introducing the text description achieves a 0.015 improvement on SSIM.

As shown in Fig. 7 (left), when provided with the language description "*pavement*", the model tends to reconstruct the scene more closely to the ground truth, whereas the baseline model randomly generates a background which distinct to the ground truth. This demonstrates the importance of semantic information for E2V reconstruction, especially in cases where semantic ambiguity exists in the event data. We also conduct an ablation study on the previous frame condition, as indicated in Tab. 3 (row 2), where performance significantly dropped across all metrics. This underscores the critical role of the previous frame condition in our diffusion pipeline, ensuring temporal consistency within the autoregressive reconstruction process.

Table 3: Ablation study on context conditions.

| Event | Text | Frame | MSE↓ | SSIM↑ | LPIPS↓ |
|-------|------|-------|------|-------|--------|
| ✓ | - | ✓ | 0.038 | 0.567 | 0.199 |
| ✓ | ✓ | - | 0.067 | 0.474 | 0.258 |
| ✓ | ✓ | ✓ | **0.023** | **0.629** | **0.194** |

**Event-guided Spatio-temporal Attention.** To demonstrate the effectiveness of our event-guided spatio-temporal attention (ESA), we conducted experiments by training the model with simple channel-wise concatenation for event input. As shown in Tab. 4, performance significantly drops without the attention mechanism, which demonstrates the effectiveness of ESA in preserving the event control on the video. Fig. 7 illustrates that our model maintains visual content close to the ground-truth, whereas the baseline method loses control over lighting and luminance. We further investigate the effectiveness of the event-aware mask loss. As is shown in Tab. 4 , the event-aware loss function achieves a clear margin of improvement compared to the baseline method (row 2). Moreover, Tab. 4 (row 3) demonstrates the effectiveness of our event-based initialization.

Table 4: Ablation study on key components."EML" denotes the event-aware mask loss. "EI" denotes event-based initialization.

| ESA | EML | EI | MSE↓ | SSIM↑ | LPIPS↓ |
|-----|-----|----|------|-------|--------|
| - | ✓ | ✓ | 0.105 | 0.468 | 0.288 |
| ✓ | - | ✓ | 0.042 | 0.443 | 0.322 |
| ✓ | ✓ | - | 0.043 | 0.482 | 0.251 |
| ✓ | ✓ | ✓ | **0.023** | **0.629** | **0.194** |

## 5    Conclusion and Future Work

**Conclusion.** In this paper, we introduce LaSe-E2V, a language-guided, semantic-aware E2V reconstruction method. Leveraging language descriptions that naturally contain abundant semantic information, LaSe-E2V explores text-conditional diffusion models with our proposed attention mechanism and loss function, thus achieving high-quality, semantic-aware E2V reconstruction. Extensive experiments demonstrate the effectiveness of our innovative framework.

**Limitations and Future Work.** Despite the promising performance of our method, LaSe-E2V has several limitations. First, the training datasets, comprising synthetic and limited real-world data, are inadequate for optimizing data-intensive diffusion models. Consequently, our method may reconstruct scenes with artifacts differing from the training data. Second, given that LaSe-E2V relies on the diffusion model, multiple denoising steps are required for high-quality videos, slowing the process. Future work can focus on accelerating the inference efficiency based on the recent progress of diffusion models [37, 9, 38, 39].

**Broader Impacts.** Based on the event camera, our LaSe-E2V enhances the capabilities of various technologies by improving the safety and robustness of intelligent systems. As this technology matures, its integration into everyday devices and systems seems likely, heralding a shift in how visual data is captured and utilized across industries.

**Acknowledgments.** This work is supported by the Guangzhou-HKUST(GZ) Joint Funding Program (No. 2024A03J0680), the Guangzhou City, University and Enterprise Joint Fund under Grant No.SL2022A03J01278, and Guangzhou Fundamental and Applied Basic Research (Grant Number: 2024A04J4072). This work is partially supported by the CCF-Tencent Rhino-Bird Open Research Fund RAGR20230120.

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

# Appendix

## A. Event voxel representation

Given a event stream $\mathcal{E}^i = \{e_{t_k}\}_{k=1}^N$ with $N$ events stream, each event $e_{t_k} \in \mathcal{E}^i$ denotes a four-element tuple $(x_k, y_k, t_k, p_k)$, reporting spatial coordinates, timestamp and polarity respectively. The event voxel representation is formulated as follows:

$$V^i(k) = \sum_j p_j \max\left(0, 1 - \left|k - \frac{t_j - t_0}{t_N - t_0}(B-1)\right|\right),\tag{8}$$

where $t_0$, $t_N$ denote the start time and end time of event stream $\mathcal{E}^i$ respectively, $k \in [0, B-1]$, $B = 5$ for our experimental setting.

## B. Additional datasets and Implementation Details

**Datasets.** For ECD, we use seven short sequences from this dataset, where the DAVIS240C [5] camera moves with 6-DOF and with increasing speed in six of them. These sequences mostly contain simple office environments with static objects. MVSEC is recorded by a synchronized stereo event camera system. Each sequence of MVSEC releases extensive ground-truth reference data for evaluations. The HQF dataset, recorded by two DAVIS240C [5] cameras, provides high-quality ground truth frames, of which the motion blur is maximally mitigated under preferable exposure. 14 sequences are contained, covering a wider range of motions and scene types, including static scenes and motion scenes of slow, medium and fast, indoor and outdoor scenes. Following the training processing, we generate language descriptions for each sequence. For a fair comparison, we select the same sequences from the three datasets as those reported in the recent benchmark of EVREAL [12].

**Implementation Details.** During training, we randomly drop input text prompts with a probability of 0.1 to enable classifier-free guidance [22]. For the reconstruction of the first clips and the accumulation error of the autoregressive pipeline, we randomly drop the first frame as the condition with 0.4 probability. During inference, we employ the DDIM sampler [56] with 50 steps and classifier-free guidance with a text guidance scale of $w = 5$ to sample videos.

**Evaluation Metrics.** The SSIM metric raises ambiguity because it involves several hyper-parameters that may differ across various codebases. For example, in the $structural\_similarity$ function of the skimage package, parameters like $gaussian\_weights$ and $sigma$ are used for spatial weighting of each patch with a Gaussian kernel, $use\_sample\_covariance$ indicate whether normalize covariances, and $K1$ and $K2$ are algorithm-specific parameters that need to be set. For this reason, we reevaluated all comparison methods by using a unified metric, denoted as the $\text{SSIM}^*$ scores.

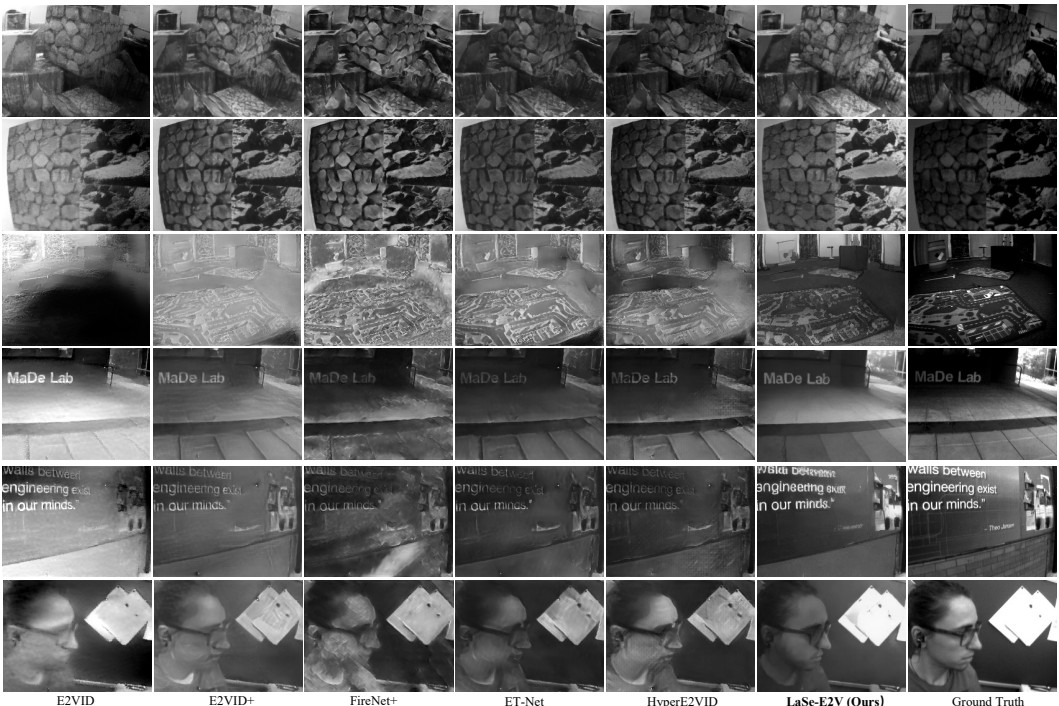

| E2VID | E2VID+ | FireNet+ | ET-Net | HyperE2VID | **LaSe-E2V (Ours)** | Ground Truth |

Figure 9: Additional qualitative results for three datasets.

## C. Additional qualitative results

In Fig. 9, we provide the additional qualitative results. For example, as shown in column 4 and 5, our method presents more accurate reconstructed results, especially for those vocabularies. In column 1, 2 and 3, our method reconstructs more details. In the supplementary material, we also provide several reconstructed video clips to demonstrate the superiority of LaSe-E2V on temporal smoothness and consistency.

## D. Complexity analysis

We provide a detailed computational complexity analysis in Tab. 5, which includes recent event-to-video reconstruction methods and related diffusion-based approaches. Our method requires a significant amount of inference time due to the 50 denoising steps, in contrast to previous single-step event-to-video methods. However, this is a common challenge for all diffusion-based models, as observed in diffusion-based super-resolution [59, 67, 65] and depth estimation methods [17, 28]. To reduce inference time, some research, *e.g.*, Instaflow [36], already explored decreasing the number of denoising steps, offering a promising direction for further improvement (as a future work) to our framework. Albeit with lower inference speed than conventional E2V methods (but higher than diffusion-based super-resolution and depth estimation methods), our work brings new ideas and may hopefully inspire new future research for event-based vision by incorporating language guidance.

## D. More analysis results

**Additional comparison methods.** We mainly compared with the latest state-of-the-art method, *e.g.*, HyperE2VID [13] in the main paper. By default, our method is superior to all the previous methods. We also provide more comparison methods in Tab. 6, which further demonstrates the superior performance of our method.

**Quantitative results on HS-ERGB dataset.** Tab. 7 provides a quantitative comparison based on three sequences (*horse_11*, *horse_12*, *horse_13*) from the HS-ERGB dataset. Existing E2V methods typically fail to reconstruct regions without events, leading to significantly worse quantitative results. Although our method may not perfectly reconstruct every detail for reality, it does generate a

Table 5: Complexity comparison on various methods. All tests are conducted on one NVIDIA Tesla 32G-V100 GPU.

| Methods | Parameters | Inference time (per frame) |
|---|---|---|
| **Conventional Event-to-Video** | | |
| ET-Net [64] | 22.18M | 0.0124 s |
| HyperE2VID [13] | 10.15M | 0.0043 s |
| **Diffusion-based Depth Estimation** | | |
| DepthFM [17] | 891M | 2.1 s |
| Marigold [28] | 948M | 5.2 s |
| **Diffusion-based Super-Resolution** | | |
| StableSR [59] | 1409M | 18.70 s |
| PASD [67] | 1900M | 6.07 s |
| SeeSR [65] | 2284M | 7.24 s |
| **Diffusion-based Event-to-Video** | | |
| LaSe-E2V (Ours) | 1801M | 1.09 s |

Table 6: Comparison with more event-to-video methods.

| Methods | ECD | | | MVSEC | | | HQF | | |
|---|---|---|---|---|---|---|---|---|---|
| | MSE↓ | SSIM↑ | LPIPS↓ | MSE↓ | SSIM↑ | LPIPS↓ | MSE↓ | SSIM↑ | LPIPS↓ |
| Zhang et al. (TPAMI2022) [74] | 0.076 | 0.519 | 0.457 | - | - | - | - | - | - |
| EVSNN (CVPR2022) [78] | 0.061 | 0.570 | 0.362 | 0.104 | 0.389 | 0.538 | 0.086 | 0.482 | 0.433 |
| PA-EVSNN (CVPR2022) [78] | 0.046 | 0.626 | 0.367 | 0.107 | **0.403** | 0.566 | 0.061 | 0.532 | 0.416 |
| CISTA-LSTC (TPAMI2023) [34] | 0.038 | 0.585 | 0.229 | - | - | - | 0.041 | 0.563 | 0.271 |
| CISTA-Flow (Arxiv2024) [35] | 0.047 | 0.586 | 0.225 | - | - | - | 0.034 | **0.590** | 0.257 |
| HyperE2VID (TIP2024) [13] | 0.033 | 0.576 | 0.212 | 0.076 | 0.315 | 0.476 | **0.031** | 0.531 | 0.257 |
| **LaSe-E2V (Ours)** | **0.023** | **0.629** | **0.194** | **0.055** | 0.342 | **0.461** | 0.034 | 0.548 | **0.254** |

reasonable output that aligns with human preference and is generally close to the distribution of the real scene. Therefore, while our results on the HS-ERGB dataset may be less significant than those on "constantly moving" datasets (ECD, MVSEC, HQF), our method is still substantially better than baseline methods.

**Impact of the prompting model.** We also tested BLIP [32] on a sampled sequence (*i.e.*, *boxes* in HQF) to further evaluate the influence of the prompting model, as shown in Tab. 8. BLIP can generate reasonable text prompts in caption-style and show nearly reconstructed performance on MSE (0.025 vs 0.027).

### E. Discussion

We begin by discussing the source of text prompts used for event data. Our framework primarily aims to establish a pipeline for reconstructing videos from events using a language-guided approach. This is based on our observation that language naturally conveys rich semantic information, which improves the semantic consistency of the reconstructed video. This form of text guidance resembles text-guided denoising [11, 47] and super-resolution [16] methods, which also use natural language as a user-friendly interface to control the image restoration process. In this work, we do not emphasize the specific sources of text prompts, as these can vary depending on the application scenario. For instance, when using a DAVIS346 camera, text can conveniently be obtained from APS frames. If a Prophesee camera is used and only event data is available, tagging-based text prompts derived from event-based multi-modality models can be employed. Additionally, in complex scenes with limited

Table 7: Quantitative comparison of HS-ERGB [58]. Results are conducted on 3 sequences with a total of 497 frames.

| Methods | MSE | SSIM | LPIPS |
|---|---|---|---|
| E2VID [50] | 0.199 | 0.382 | 0.736 |
| HyperE2VID [13] | 0.161 | 0.374 | 0.745 |
| **LaSe-E2V (Ours)** | **0.078** | **0.429** | **0.665** |

Table 8: Comparisons between different prompting models on *boxes* of HQF.

| Prompting Models | MSE↓ | SSIM↑ | LPIPS↓ |
|---|---|---|---|
| RAM [73] | 0.025 | 0.557 | 0.196 |
| BLIP [32] | 0.027 | 0.546 | 0.207 |

event data, human intervention can be incorporated interactively to enable user control. Overall, our method offers a flexible and generic language-guided interface for E2V reconstruction.

Regarding the role of text information, it serves as a crucial component to activate the semantic prior in the diffusion model and compensate for missing semantic content. In regions with sufficient event data, our approach enhances reconstruction performance by leveraging the semantic priors provided by the text prompts, ensuring high fidelity. For regions with sparse event data, the method relies solely on the text prompts to reconstruct scenes that align with human expectations. While this approach may introduce textures or details that differ from the actual ones, it still produces images closer to reality than previous methods. In contrast, prior E2V models typically reconstruct such regions as indistinct haze, far from the true distribution of real images, as shown in Fig. 1. The quantitative comparison in Tab. 7 further demonstrates the superiority of our approach.

Admittedly, hallucination is a potential side effect introduced by incorporating semantic priors from the diffusion model into our framework. While our proposed techniques effectively reduce hallucinations, completely eliminating them remains a challenge in the diffusion model research community. Extensive experiments (Tab. 1 and Fig. 3) show that our method significantly improves fidelity and reduces hallucinations in scenes with adequate event data. Additionally, our approach exhibits superior reconstruction performance even in scenes with insufficient event data (Tab. 7). It is important to note that eliminating hallucinations entirely for regions with sparse event data is not feasible. However, our method effectively leverages the semantic priors in the diffusion model to produce images closer to the true scene distribution, whereas previous E2V methods (e.g., HyperE2VID) fail in these regions, as demonstrated in Fig. 1.

For future work, given that our method reliably ensures fidelity in regions with sufficient event data, it is practical to assign a confidence map based on event density to identify high-confidence regions. For safety-critical applications, decisions could be made based on both the reconstructed video and the confidence map, allowing simultaneous consideration of image quality and safety. Moreover, event-based multi-modality models offer a potential source of semantic information, which could be explored in future research. Since these models are trained with large-scale event-image paired data, they have the potential to provide prior semantic information absent in event data.

