# OpenReview forum: "LaSe-E2V: Towards Language-guided Semantic-aware Event-to-Video Reconstruction"
_NeurIPS.cc/2024/Conference — NeurIPS 2024 poster_

### Official Review · Reviewer_uWG8 · 2024-07-07

**Soundness:** 3
**Presentation:** 3
**Contribution:** 3
**Rating:** 6
**Confidence:** 3

**Summary:**

This paper proposes a language-guided event-based video reconstruction method LaSe-E2V, which introduces the popular language model into event-based imaging tasks. The LaSe-E2V is generally based on a diffusion model. In order to further improve the method, this paper proposes a series of designs, such as event-guided spatio / temporal attention for the fusion between event and previous reconstructed frames, previous frame conditioning for ensuring consistency, event-aware mask loss, and event-aware noise initialization. Experimental results demonstrate the effectiveness of the proposed methods.

**Strengths:**

1. This paper introduces the popular language model into the event-based imaging community, providing new ideas for subsequent works.
2. The event-guided spatio / temporal attention provides a new idea for event-image fusion.
3. The event-aware mask loss and the event-aware noise initialization are specially designed for event-based imaging based on diffusion models.

**Weaknesses:**

There are some unclear points, please see the Questions part.

**Questions:**

1. The proposed method requires a description of scenes contained in the event streams to be reconstructed. What if the description of the scene is unknown (For example, a ``blind'' event stream)? Can existing models handle this situation?
2. The running times and GPU memory costs for inference are absent. Diffusion models are still new for the event-based imaging community, and offering these values can help readers better understand the proposed method and do subsequent work.
3. It says in Line 228 of the paper that the definition of the SSIM metric has ambiguity. What's the ambiguity?
4. In the 3rd column of Fig. 4, there are more artifacts in the text part reconstructed by the proposed method than other methods. It's an interesting phenomenon, and it would be better if some explanations were given.
5. There is a typo ``evet'' in the Line 66 of the manuscript.

**Limitations:**

Please see the Questions part.

---

> ### Author Rebuttal · Authors · 2024-08-07
>
> Thank the reviewer for the constructive comments and valuable concerns.
>
> > **Can the model handle the situation when the description of the scene is unknown?**
> >
> - Yes, our model can handle such scenarios. If the text description is unavailable, our framework defaults to a conventional event-to-video approach. As demonstrated in Tab. 2 (row 1), even without text input, our method **still achieves reasonable performance**.
> - Additionally, event-based multi-modality models such as EventBind[77] and EventClip[78] can also be employed to **generate text descriptions directly from the event stream**.
>
> > **About running times and GPU memory costs for inference.**
> >
> - Thanks for the question. We have included a complexity comparison in **Global Response to All Reviewers.** As illustrated in Tab. A-1, our method requires significant inference time due to the multiple denoising steps, showing a known limitation of diffusion models. However, while diffusion models have been employed in tasks like super-resolution and depth estimation, “*they are still new to the event-based imaging community*”(***uWG8***). As such, our approach could hopefully *“provide new ideas for subsequent works”(**uWG8**)* in this field.
>
> > **Ambiguity of SSIM metric.**
> >
> - The SSIM metric raises ambiguity because it involves several hyper-parameters that may differ across various codebases. For example, in the `structural_similarity` function of the **skimage** package, parameters like `gaussian_weights` and `sigma` are used for spatial weighting of each patch with a Gaussian kernel, `use_sample_covariance` indicates whether to normalize covariances, and `K1` and `K2` are algorithm-specific parameters that need to be set.
>
> > **About explanations for the artifact of text reconstruction in Fig.4.**
> >
> - Thanks for the insightful question. Although our method achieves superior overall performance, some artifacts persist in the text part reconstruction. This issue arises because we depend on the prior of the pre-trained diffusion model (SD2 [42]), which also faces challenges in text generation. However, the recently released SD3 [79] claims to show improved text rendering capabilities, which could potentially address this problem.
>
> > **Typo: "evet".**
> >
> - Thanks for pointing out the typo. We will correct it in the final version.
>
> **Additional Reference:**
>
> [77] Zhou J, Zheng X, Lyu Y, et al. E-clip: Towards label-efficient event-based open-world understanding by clip[J]. arXiv, 2023.
>
> [78] Wu Z, Liu X, Gilitschenski I. Eventclip: Adapting clip for event-based object recognition[J]. arXiv, 2023.
>
> [79] Esser P, Kulal S, Blattmann A, et al. Scaling rectified flow transformers for high-resolution image synthesis[C]//ICML. 2024.

---

> > ### Comment · Reviewer_uWG8 · 2024-08-12
> >
> > Thanks for your rebuttal. I'll keep my rating.

---

### Official Review · Reviewer_H8fU · 2024-07-10

**Soundness:** 3
**Presentation:** 3
**Contribution:** 2
**Rating:** 6
**Confidence:** 4

**Summary:**

The paper explores the use of Denoising Diffusion Models (DDM) for event-based video reconstruction task. The most important contribution, in my opinion, is the improvement in video quality. The researchers adapted an existing model (ModelScope Text-to-Video) for the  event-based video reconstruction task. They introduced ESA, that modiffies a conditional inpainting technique (presented on High-Resolution Image Synthesis with Latent Diffusion Models). Additionally, they used text conditioning to further enhance video quality.

The innovative contributions of the paper are two. The fist one is the Event-aware Noise Initialization; This technique enables the use of frame I_(t-1) to reconstruct frame I_(t) during the inference stage (autoregressive).
The second one is the Event-aware Mask Loss; This new loss function is designed to improve the temporal consistensy.

In summary, the contributions are:

1. An existing text-to-video model (diffusion-based) was adapted for the event-based recosntruction task.
2. The model was adapted to event data using a conditional inpainting technique. Also proposed Event-aware Noise Initialization and Mask Loss to improve video quality.
3. The result is a new state-of-the-art in event-to-video reconstruction.

However, the proposed model has several drawbacks. The DDM "hallucinates" content, especially in areas with low or no event data, which can be problematic for applications like object detection in the context of self-driving cars.

**Strengths:**

As mentioned before, the main contribution of this paper is the improvement in video quality. Also, this paper proves that it is possible to use diffusion-based models for the event-based video reconstruction task.

**Weaknesses:**

Diffusion-based models tend to "hallucinate," producing some parts on a frames that are far from reality (in the event-based video reconstruction task), which can be problematic for applications like object detection in the context of safety (self-driving cars).

Another problem is that the proposed model uses text prompts for video generation. Although these text prompts allow some control over the content, they also introduce a lot of ambiguity. This leads to a trial-and-error process (prompt engineering) until the most realistic reconstruction is achieved according to the user.

Additionally, the proposed model requires very high computational resources. Inference times are not specified in the paper, but it is known that the model does not run in real-time.

**Questions:**

1) Although traditional metrics (LPIPS, MSE, and SSIM) show an increase in video reconstruction quality, the qualitative results, especially in Figure 3, reveal that the reconstructed frames do not reflect reality in specific areas. These artifacts represent a smaller percentage of the image (in terms of pixels) and are not captured by traditional metrics, how does this affect the overall video quality?

2) In the inference stage, no mention is made of the length (number of frames) that the model can reconstruct. If there is a limit on the number of frames (how many frames and) , how is temporal consistency between reconstructed sections resolved?

3) Despite discussing the model's temporal consistency, no metrics verify this.

4) On line 134, the variable V is introduced, representing the conversion of event data to voxels. However, it is unclear whether V represents a segment with N temporal bins or N segments (the assumption is that V represents N segments). Clarification of this detail is necessary.

5) On line 139, the input latent representation is introduced through the variable \hat{\epsilon} . In DDM, \epsilon variable represents noise, and the noisy input latent representation normaly is denoted with z_t in this case as \hat{z}^{i}_{t}, since it is the latent image after applying the noise and the representation of the events z^{i}_{e}. I'm not sure if it's correct.

6) The Event-guided Spatio-temporal Attention (ESA) module is very similar to the technique used in SD (High-Resolution Image Synthesis with Latent Diffusion Models) for inpainting. A reference to this would be beneficial.

7) In section 4.2, on line 162, the title "Event Spatial Attention" mentions a cross-attention technique. However, the title does not reflect this, causing confusion. It could be changed to "Event Spatial Cross-Attention". Similarly, on line 176, the title "Event Temporal Attention" maybe should be changed to "Event Temporal Cross-Attention."

7) In equation 8, line 199, the value of \(\lambda\) is not mentioned.

8) No mention is made of the computational cost in the inference stage, nor is the inference time mentioned. Could these data be mentioned?

**Limitations:**

Due to the DDM's tendency to hallucinate, the model cannot be used in self-driving cars or other computer vision applications where safety is involved.

It is believed that the model cannot be run in real time, much less on an embedded device.

---

> ### Author Rebuttal · Authors · 2024-08-07
>
> Thank the reviewer for appreciating our work with valuable suggestions. We address the comments below.
>
> > **About the "hallucination" issue of diffusion-based models.**
> >
> - Yes, this is indeed a problem for the diffusion models. To address this, we have proposed novel techniques (i.e., ESA module, Event-aware Mask Loss, and Event-aware Noise Initialization) to mitigate the diversity objectives of the diffusion model. The proposed techniques thereby enhance the consistency between events and the video and reduce the hallucination issue.
> - Please be noted that, in static areas without event data, the model tends to reconstruct these regions with hallucination. By incorporating text descriptions, the hallucination issue can be mitigated to some extent, as shown in Fig. 7 (left).
>
> > **Text could introduce ambiguity and lead to a rial-and-error process.**
> >
> - Unlike existing image synthesis pipelines, our method primarily relies on event data, with text descriptions serving as the **supplementary guidance** only when events are too sparse or not trigged. This approach minimizes the potential for ambiguity.
> - Additionally, our method requires only **coarse** text descriptions to effectively leverage the semantic prior in the T2I diffusion model. Most of these descriptions simply identify objects, such as "car" or "road," generated by the tagging model RAM, which introduces little ambiguity. As shown in Tab. 1 and Fig. 4, our experiments confirm that this coarse-grained text is effective and does **not** necessitate a trial-and-error process.
>
> > **Computational Resources and Inference Times.**
> >
> - Thanks for the suggestion. We have included a complexity comparison in **Global Response to All Reviewer*s***. As shown in Tab. A-1, our method requires considerable inference time due to multiple denoising steps, showing a limitation of diffusion models. However, our approach “*could inspire future work in the field*”(***ueNs***) by incorporating language guidance and proves “*it is possible to use diffusion-based models for the event-based video reconstruction task*”(***H8fU***).
>
> > **The artifacts in Fig. 3 are not captured by traditional metrics in Tab. 1.**
> >
> - The reviewer might misunderstand Fig.3. Fig. 4 presents the qualitative results corresponding to the quantitative data in Tab. 1, derived from the widely-used ECD, MVSEC, and HQF datasets. These results demonstrate a significant improvement in reducing artifacts. Conversely, Fig. 3 illustrates qualitative results from a **different** dataset (HS-ERGB) that involves fast motion.
> - Additionally, the traditional metrics (MSE, SSIM, and LPIPS) used in Tab. 1 are widely recognized in recent E2V research. These metrics have been proven to effectively assess results at the pixel level (MSE), structure level (SSIM), and in terms of human perception (LPIPS).
>
> > **How is temporal consistency between reconstructed sections resolved?**
> >
> - As mentioned in Line 184, we condition the **frame from the previous section** to enable an **auto-regressive** pipeline for long video reconstruction. The ablation study in Tab. 2 (2nd row vs. 3rd row) has demonstrated the effectiveness of the previous frame condition to ensure the temporal consistency between sections. Theoretically, the auto-regressive pipeline imposes **no limitations** on the number of frames.
>
> > **Metrics for temporal consistency.**
> >
> - Upon the suggestion, we evaluate the results based on the temporal quality metrics from VBench [76]. As shown in Tab. A-4, our method significantly outperforms others on subject consistency and background consistency, while achieving comparable performance on motion smoothness. These results demonstrate the effectiveness of our approach in maintaining temporal consistency. We will include these results in the camera-ready version.
>
> Table A-4: Quantitative Comparison on temporal consistency on ECD based on VBench metrics.
>
> |  | Subject Consistency | Background Consistency | Motion Smoothness |
> | --- | --- | --- | --- |
> | E2VID [40] | 52.14% | 85.26% | 97.62% |
> | FireNet [44] | 49.61% | 82.78% | 98.40% |
> | E2VID+ [48] | 51.83% | 85.33% | 97.56% |
> | FireNet+ [48] | 47.97% | 83.23% | 97.11% |
> | SPADE-E2VID [7] | 50.58% | 82.61% | **98.41%** |
> | SSL-E2VID [37] | 51.89% | 84.86% | 95.97% |
> | ET-Net [54] | 55.49% | 86.85% | 97.72% |
> | HyperE2VID [12] | 50.41% | 83.50% | 97.59% |
> | **LaSe-E2V (Ours)** | **84.25%** | **93.39%** | 98.11% |
> | *GT (Empirical Max)* | *88.29%* | *93.65%* | *98.67%* |
>
> > **Clarification on the Variable "V".**
> >
> - Thanks for the question. The **V** in bold represents N segments. We will clarify this in the final version.
>
> > **Clarification on the Variable $\epsilon$.**
> >
> - Yes, the noisy input latent representation is typically denoted as $z_t$ during the training process, while $\epsilon$ refers to the sampling noise input during inference. Sorry for the confusion. We will clarify it in the final version.
>
> > **Adding reference to ESA.**
> >
> - Thanks for the suggestion. We will add the reference for this module in the revision. While SD [42] serves as a baseline architecture to incorporate text embedding, we introduce ESA to enhance spatial-temporal consistency between the event data and video by considering the unique characteristics of event data.
>
> > **Adding Cross-Attention to section title of ESA.**
> >
> - Thanks for the insightful comment. We will revise the title in the final version to better reflect the inclusion of cross-attention.
>
> > **About the value of $\lambda$.**
> >
> - Thanks for the question. The value of $\lambda$ is set to 0.01 for all the experiments. We will clarify it in the implementation details.
>
> **Additional Reference:**
>
> [76] Huang Z, He Y, Yu J, et al. Vbench: Comprehensive benchmark suite for video generative models[C]/CVPR. 2024.

---

> > ### Comment · Reviewer_H8fU · 2024-08-12
> >
> > Hallucinations:
> >
> > I understand that the ESA module and mask loss help mitigate the problem of "hallucinations." However, this information does not present anything new regarding hallucinations. Additionally, as shown in Figure 7, a prompt had to be manually adjusted to generate a similar background, but it still differs from reality.
> >
> >
> > Ambiguity in Text Prompts:
> >
> > As shown in Figure 7, when no events are triggered in large regions, it becomes necessary to manually generate a text prompt that, according to the user, matches reality. This introduces ambiguity into the process. Furthermore, even when a close image can be generated with the prompt, the textures and forms often differ significantly from the real ones.
> >
> > Computational Resources and Inference Times:
> >
> > Table A-1 clarifies this question; however, it would be helpful to include the memory (VRAM) required to run each model.
> >
> >
> > Artifacts Not Captured by Traditional Metrics in Figure 3:
> >
> > Figure 3 shows the qualitative results from the HS-ERGB dataset, but no quantitative results are provided for this dataset. The ECD, MVSEC, and HQF datasets might not capture the artifacts (hallucinations) produced by the proposed LaSe-E2V model because the camera is constantly moving in those scenarios. However, in situations like the HS-ERGB dataset, where both the background and the camera are stationary, the proposed model may yield worse quantitative results for the HS-ERGB dataset.
> >
> > Additionally, if we look at Figure 4, in the third column, there is a poster with letters that are completely distorted by the proposed LaSe-E2V model, making it impossible to read the content. However, since these are small regions relative to the size of the image, traditional metrics (MSE, SSIM, and LPIPS) do not adequately reflect these artifacts.
> >
> >
> > Temporal Consistency Between Reconstructed Sections:
> >
> > Information about the autoregressive pipeline helps clarify doubts related to video generation during the inference stage.
> >
> >
> > Metrics for Temporal Consistency:
> >
> > Doubts related to temporal consistency have been clarified.
> >
> >
> > Reference to ESA:
> >
> > References regarding the ESA module and the main similarities and differences with respect to SD (in inpainting) were not added.
> >
> >
> > Remaining Questions:
> >
> > The rest of the questions were answered satisfactorily.

---

> ### Author Response · Authors · 2024-08-13
> **Author response to Reviewer H8fU (1/2)**
>
> Thank the reviewer for elaborating on the points. We address the reviewer's concerns below:
>
> > **Hallucinations**
> >
> - Hallucination is a potential **side effect** introduced by incorporating semantic priors from the diffusion model into our framework. In this regard, our proposed techniques are demonstrated to be effective in reducing hallucinations, but **not entirely** eliminating them, which is still a challenge in the diffusion model research community. Extensive experimental results (Tab. 1 and Fig. 4) show that our method significantly **improves fidelity** and reduces hallucinations in the scenes with sufficient event data.  Our method also exhibits better reconstruction performance for the scenes even with insufficient event data (Tab. A-6). Please be reminded that it is impossible to totally remove the hallucination effect for regions with insufficient events. However, our method indeed effectively leverages the semantic prior in the diffusion model to reconstruct the image more **closer to the distribution** to the real scene, whereas the previous E2V methods (e.g., HyperE2VID) **fail** in these regions, as shown Fig. 1.
> - Our framework mainly focuses on providing a potential pipeline to reconstruct the video from events with the guidance of language. This is based on our finding that language naturally conveys abundant semantic information, which is beneficial in enhancing the **semantic consistency** for the reconstructed video (see Lines 39-41). The text guidance can serve as a form of human intervention through text prompts, which is **similar** to the way of the text-guided denoising [80,81] and super-resolution[82] methods.
> - As shown in Fig. 1, it compares the difference between our method and the previous E2V method (i.e., HyperE2VID). For region with insufficient events, although our results still differs from reality in some details, they indeed reconstructs a scene according to human preference. In contrast, the previous E2V method always reconstructs **haze** in these regions. For Fig. 7, we will update the results from previous E2V methods for clearer comparison. Quantitatively, Tab. A-6 also shows the superiority of our method in the scene with insufficient events.
>
> > **Ambiguity in Text Prompts**
> >
> - Our framework incorporates text descriptions as complementary information, marking the first instance of allowing language guidance in the E2V pipeline. This approach is believed to "*inspire future work*" (***ueNs***) and "*provide new ideas*" (***uWG8***). While this may introduce some ambiguity, it also offers the flexibility to manually adapt the reconstructed video according to user preferences, as demonstrated in Fig. 8. The text guidance manner is also similar to the way of the text-guided denoising [80,81] and super-resolution[82] methods.
> - In regions with sufficient event data, our method improves reconstruction performance by leveraging the semantic priors from the text prompts, ensuring high fidelity. For regions with insufficient event data, the method relies solely on the text prompts to reconstruct a scene that aligns with human preferences. Although this approach may produce textures or details that differ from the real ones, it still generates images that are closer to reality. In contrast, the previous E2V models always reconstruct these regions as indistinct haze, far from the true distribution of real images. Fig. 1 shows the qualitative comparison. Tab. A-6 also shows the quantitative comparison and demonstrates the superiority of our method. We will add more comparisons on these scenes with insufficient events in the revision.
>
> > **Computational Resources and Inference Times**
> >
> - Table A-5 provides the GPU memory for different E2V models. It shows similar memory costs among different methods.
>
> Table A-5: GPU memory cost of different E2V methods. All tests are conducted on one NVIDIA Tesla 32G-V100 GPU.
>
> | Methods | GPU memory |
> | --- | --- |
> | E2VID [40] | 12120 |
> | ET-Net [54] | 12218 |
> | HyperE2VID [12] | 12227 |
> | LaSe-E2V (Ours) | 12139 |

---

> > ### Comment · Reviewer_H8fU · 2024-08-14
> >
> > Hallucinations
> >
> > While this paper demonstrates improved video reconstruction quality compared to previous works (such as HyperE2VID), it is important to address the side effect of "hallucinations" in the limitations section. These include small artifacts that traditional metrics like LPIPS, MSE, and SSIM cannot capture.
> >
> > Ambiguity in Text Prompts
> >
> > Similar to hallucinations, the potential limitations of introducing "ambiguity" with text prompts in the reconstruction pipeline should be discussed. These effects could lead to unsuitable video reconstructions for safety-critical applications, such as self-driving cars, due to the risk of generating nonexistent objects. This concern should also be highlighted in areas like computational photography, where fidelity is crucial.
> >
> > Computational Resources and Inference Times
> >
> > Table A-5 does not specify the units for memory requirements—are they in gigabytes or megabytes? Additionally, what precision is being used—float16, float32, or something else? Moreover, the significant differences in the number of parameters between models do not seem to align with their reported memory consumption. For instance, LaSe-E2V is reported to have 1,801 million parameters, while HyperE2VID has 10.15 million. Yet, LaSe-E2V’s memory consumption (12,139 unknown units) is listed as less than HyperE2VID’s (12,227 unknown units). Please address and correct these discrepancies.

---

> > > ### Author Response · Authors · 2024-08-14
> > > **Author response to Reviewer H8fU**
> > >
> > > Thank the reviewer for the insightful discussion! We will include these points in the camera-ready version.
> > >
> > > > **Hallucinations**
> > > >
> > > - We will manually identify small artifacts that traditional metrics fail to capture. These findings will be discussed in the Limitations section of the camera-ready version.
> > >
> > > > **Ambiguity in Text Prompts**
> > > >
> > > - Considering that our method reliably ensures fidelity in regions with sufficient event data, it is practical to assign a **confidence map** based on event density to identify high-confidence regions. For safety-critical applications, it is feasible to make decisions based on both the reconstructed video and the confidence map. This approach allows for simultaneous consideration of image quality and safety. We will include this limitation and the potential solution in the revision.
> > >
> > > > **Computational Resources and Inference Times**
> > > >
> > > - Sorry for the confusion. The unit in the Tab. A-5 is **megabytes** (MB). **Float32** is used for all the methods.
> > > - Previous E2V methods are based on a recurrent architecture. Theoretically, recurrent models require more memory because they initially allocate enough memory to store the previous states, which potentially enhances the inference speed. However, when we clear the pre-allocated memory before each iteration, we observe that memory usage for HyperE2VID drops to 1372MB. There is also a minor decrease in inference speed due to the time required to reallocate memory. This represents a trade-off between memory usage and time cost, potentially influenced by CUDA tools. We will clarify this in the revision.

---

> > > > ### Comment · Reviewer_H8fU · 2024-08-14
> > > >
> > > > Thank you to the authors for the clarifications and the additional experiments. After thorough discussion, all my doubts and concerns have been addressed. My rating for this work is 6.

---

> ### Author Response · Authors · 2024-08-13
> **Author response to Reviewer H8fU (2/2)**
>
> > **Artifacts Not Captured by Traditional Metrics in Figure 3**
> >
> - Tab. A-6 provides a quantitative comparison based on three sequences (*horse_11*, *horse_12*, *horse_13*) from the HS-ERGB dataset. Existing E2V methods typically fail to reconstruct regions without events, leading to significantly **worse** quantitative results. Although our method may **not perfectly** reconstruct every detail for reality, it does generate a **reasonable** output that aligns with human preference and is generally **close to the distribution** of the real scene. Therefore, while our results on the HS-ERGB dataset may be less significant than those on "constantly moving" datasets (ECD, MVSEC, HQF), our method is still **substantially better** than baseline methods.
> - Regarding the artifacts in the letters, as also noted by reviewer ***uWG8***, these issues arise because our method relies on the pre-trained diffusion model (SD2 [42]), which also struggles with text rendering. However, the recently released SD3 [79] claims to have improved text rendering capabilities, which could potentially address this problem and improve the performance. We believe that the MSE can capture these artifacts at **pixel level**. However, although minor artifact in small regions exists, our method excels in **overall** image quality, since previous methods tend to reconstruct misty areas in front of the poster in Fig. 4 (3rd column)
>
> Table A-6: Quantitative comparison of HS-ERGB. Results are conducted on 3 sequences with a total 497 frames.
>
> |  Methods |  | HS-ERGB |  |
> | --- | --- | --- | --- |
> |  | MSE | SSIM | LPIPS |
> | E2VID | 0.199 | 0.382 | 0.736 |
> | HyperE2VID | 0.161 | 0.374 | 0.745 |
> | **LaSe-E2V (Ours)** | **0.078** | **0.429** | **0.665** |
>
> > **Reference to ESA**
> >
> - Our ESA module is specifically designed to enhance spatio-temporal consistency between events and videos. In contrast, the original SD [42] serves as a baseline attention mechanism for integrating conditional input. Our approach differs in **attention design**. While the original SD relies solely on **simple cross-attention** to incorporate various feature conditions, our ESA module takes into account the unique spatial and temporal characteristics of event data. It introduces two **distinct attention** mechanisms respective to the **spatial** domain and **temporal** domain, which fully leverage the constraints provided by the event data and ensure spatio-temporal consistency.
>
> **Additional Reference:**
>
> [80] Duan H, Min X, Wu S, et al. UniProcessor: A Text-induced Unified Low-level Image Processor[C]//ECCV. 2024.
>
> [81] Qi C, Tu Z, Ye K, et al. Tip: Text-driven image processing with semantic and restoration instructions[C]//ECCV. 2024.
>
> [82] Gandikota K V, Chandramouli P. Text-guided Explorable Image Super-resolution[C]//CVPR 2024.

---

### Official Review · Reviewer_ueNs · 2024-07-12

**Soundness:** 3
**Presentation:** 3
**Contribution:** 3
**Rating:** 5
**Confidence:** 5

**Summary:**

This paper addresses the issue of artifacts and regional blur in existing event-to-video (E2V) reconstruction algorithms by leveraging the rich semantic information in language to enhance the semantic consistency of reconstructed videos. The authors propose a language-guided E2V generation model that employs existing text-conditional diffusion models as a framework. They use an Event-guided Spatiotemporal Attention (ESA) module for fine-grained spatial alignment and temporal continuity, an event-aware mask loss for further ensuring temporal consistency, and an event-aware noise initialization to address training-testing discrepancies. Extensive comparative experiments validate the algorithm's performance, and ablation studies demonstrate the effectiveness of each component.

**Strengths:**

1 This paper is the first to tackle the event data reconstruction task from a language-guided perspective. This approach could inspire future work in the field.

2 The proposed algorithm achieves optimal performance on nearly all metrics across multiple datasets, demonstrating superior visual effects and validating the algorithm's effectiveness.

3  The paper presents a high-quality body of work, including effective methods and extensive experimental validation.

**Weaknesses:**

1 The paper lacks a detailed comparison of the algorithm's inference speed and model size. The authors acknowledge the inherent speed limitations of using diffusion models, emphasizing the importance of providing this comparison.

2 In practical applications lacking APS reference frames, obtaining accurate textual information that matches the scene description is difficult or nearly impossible. Using existing text generation models to extract semantic information introduces additional reference, compromising fairness to some extent.

**Questions:**

Why is it necessary for the event activation regions of adjacent frames to be similar, rather than the regions without events?

**Limitations:**

The authors discuss the limitations of their work, particularly concerning the required training data and the inference speed of diffusion models, in accordance with official guidelines.

---

> ### Author Rebuttal · Authors · 2024-08-07
>
> Thank the reviewer for the valuable suggestions. We address the questions below.
>
> > **Detailed comparison of inference speed and model size.**
> >
> - Please refer to the **Global R*esponse to All Reviewers***. As shown in Tab. A-1, our method requires considerable inference time due to multiple denoising steps, showing a limitation of diffusion models. However, our approach “*could inspire future work in the field*”(***ueNs***) by incorporating language guidance and proves “*it is possible to use diffusion-based models for the event-based video reconstruction task*”(***H8fU***).
>
> > **Textual information is difficult to obtain and compromises fairness.**
> >
> - We respectfully disagree. The textual description is not difficult to obtain for our framework. Please note that our method only requires coarse text descriptions to leverage the capabilities of the pre-trained T2I diffusion model. These descriptions, generated by the tagging model RAM, simply identify objects in a tagging style, such as "car," "road," and "tree". These can be easily provided by humans in the absence of APS frames.
> - Also, please note that it is not uncommon to introduce additional priors for event-to-video (E2V) reconstruction task. For instance, Zhang et al. [72] incorporated optical flow to address event-to-image reconstruction as a linear inverse problem. Our work is the first to introduce text as a guiding prior for event-to-video reconstruction, which, as noted by reviewer ***ueNs***, *"could inspire future work in the field".*
>
> > **Why is it necessary for the event activation regions of adjacent frames to be similar?**
> >
> - Sorry for the misunderstanding. It should be the regions without events to be similar. The value zero is assigned to the event-activated area, and vice versa. We will rectify Line 196 in the version.

---

> > ### Comment · Reviewer_ueNs · 2024-08-12
> >
> > Thank you for your response. My concern still lies with the acquisition of the text. All reviewers mentioned that the issue of hallucinations makes reconstruction unreliable. Event cameras are not good at capturing image details but are instead suitable for high-speed, real-time applications, unreliable reconstruction contradicts the motivation for using event cameras. Additionally, the reconstruction method in this paper is nearly 100 times slower than previous event reconstruction algorithms. If we consider the acquisition of text prompts, the application scenarios become even more limited.
> >
> > If we do not consider the application and focus solely on the method itself, the paper does not explore the impact of different text prompts on the results. The authors mention that "our method requires only coarse text descriptions" and that "this coarse-grained text is effective and does not necessitate a trial-and-error process." However, if the text information is insignificant, then what is the significance of the text in this model? Is it the case that relying solely on a pre-trained diffusion model would already yield good results? This also contradicts the motivation of the paper.
> >
> > After reading the concerns raised by other reviewers, I found the text-related parts of the experiments confusing. The qualitative results do not indicate the text used, making it difficult to assess whether the improvements in the results are related to the text. For instance, in Figure 4, the first three scenes show only an improvement in contrast, making it hard to see any other differences. In the second-to-last column, the letter "g" is clearly reconstructed incorrectly. What caused this? It's impossible to determine whether this is related to the text prompt.
> >
> > Regarding the source of the text prompts for the events, the authors mentioned in their response to me that "These can be easily provided by humans in the absence of APS frames." However, in their response to another reviewer, they stated that "event-based multi-modality models such as EventBind[77] and EventClip[78] can also be employed to generate text descriptions directly from the event stream." For the former, I cannot imagine whether a person would be watching the events to provide a text description or watching the scene to do so. It seems that if a person can directly observe the scene, then reconstruction wouldn't be necessary. If a person is observing the events, the sparsity of the events makes it difficult to provide an accurate description. For the latter, if event-based multi-modality models are used, these models would require the presence of events to function. However, the authors emphasize that the text is meant to enhance areas without events. Can these models provide accurate descriptions for sparse event regions in the absence of events?
> >
> > As this paper focused on event reconstruction with text as additional guidance, I believe it is essential to thoroughly explore the motivation and role of the text.

---

> > > ### Author Response · Authors · 2024-08-13
> > > **Author response to Reviewer ueNs (2/2)**
> > >
> > > > **Regarding the source of the text prompts for the events**
> > > >
> > > - Our framework primarily focuses on providing a potential pipeline for reconstructing videos from events with a **language-guided perspective**. This is based on our finding that language naturally conveys abundant semantic information, which enhances the semantic consistency of the reconstructed video (see Lines 39-41). Such text guidance manner is **similar** to text-guided denoising [80,81] and super-resolution [82] methods, which also rely on natural language as a user-friendly interface to control the image restoration process. In this work, we do not focus much on the specific sources of the text prompts, as these can be flexible and generic, varying depending on the application scenarios. For example, if a DAVIS346 camera is used, obtaining text from APS frames is convenient. If a Prophesee camera is used and only event data is available, it is feasible to provide tagging-based text prompts derived from event-based multi-modality models. Additionally, in complex scenes lacking sufficient event data, it is practical to incorporate human intervention in an interactive manner to support human control. Overall, our method provides a language-guided interface for E2V reconstruction that could be **feasible** and **generic**.
> > > - The reviewer might misunderstand the significance of the text prompts. The text is **not only** used to reconstruct regions without events. For regions with event data, our method incorporates text to provide complementary semantic information, ensuring semantic consistency and further improving performance.  For regions without event data, our method can rely on the text to reconstruct a reasonable scene close to the distribution of the real images. The results in Tab. 1 and Tab A-6 have demonstrated the superiority of our method in both scenarios.
> > > - Regarding event-based multi-modality models, they offer an option for providing semantic information, which can be explored in future work. Since these models are trained with large-scale event-image paired data, they have the potential to provide prior semantic information that is absent in event data. Based on these text prompts, the method reconstructs video from event data, while effectively exploiting the semantic prior in the diffusion model to ensure semantic consistency.
> > >
> > > Thanks for the insightful question! We will include this discussion in the camera-ready version.

---

> ### Author Response · Authors · 2024-08-13
> **Author response to Reviewer ueNs (1/2)**
>
> Thanks for bringing up this thoughtful discussion.
>
> > **About the hallucinations issue**
> >
> - We respectfully disagree with the reviewer. Please note that all reconstructed results are **not unreliable**. In regions with sufficient event data, our method enhances performance with text guidance while ensuring fidelity. As demonstrated in Tab. 1 and Fig. 4, our method significantly **outperforms** previous methods. We also encourage the reviewer to recheck the demo video provided in our supplementary materials. For regions with insufficient event data, our method relies primarily on text prompts to reconstruct a scene that aligns with human preferences. As demonstrated in Fig. 1, HyperE2VID struggles with **severe artifacts** (haze-like artifacts), typically in the background regions, far from the true distribution of real images. Whereas, our method exhibits **higher-quality** reconstruction results that are closer to the ground truth. The results indicate that our method can subtly leverage the semantic information from language and thus ensure semantic consistency of the reconstructed video. To further demonstrate the effectiveness of our method, we provide a quantitative comparison of the HS-ERGB dataset, which includes larger regions lacking event data, as shown in Tab. A-6. Our method substantially **outperforms** baseline methods with an MSE of 0.083. Apparently, the quantitative results verify the superiority and reliability of our method.
> - In this work, we mainly focus on a new research direction of exploring E2V reconstruction from a language-guided perspective, as affirmed by other reviewers: "This approach is expected to "*inspire future work*" (***ueNs***) and "*provide new ideas*" (***uWG8***). We did not focus much on the computation efficiency of the diffusion model, which will be left as a future work. Please be noted that using language guidance does not actually hamper the application value. Such a text-guided manner is also similar to recent methods used in text-guided denoising [80,81] and super-resolution [82], which have provided broad applications for low-level vision in a user-friendly manner. Our research is **application-significant** as the language provides abundant semantic information, beneficial for ensuring the **semantic consistency** of the reconstructed video. The text prompts also provide a way of human intervention to control the reconstruction process.
>
> > **What is the significance of the text in this model?**
> >
> - The text information serves as a **crucial** component to **activate** the semantic prior in the diffusion model. As clarified above, our method effectively enhances performance in regions **both** with sufficient and insufficient event data by utilizing text prompts. To simplify text prompts and enhance the model's robustness, we utilize only coarse, tagging-style descriptions for training. Without text prompts, it is challenging to exploit the semantic prior in the diffusion model to ensure semantic consistency. As shown in Tab. 3 (1st row), without the text prompt, our method yields modest results. However, when text information is incorporated, it significantly **improves** the model's ability to leverage the semantic prior in the diffusion model, thereby enhancing performance.
>
> > **Text-related parts of the experiments**
> >
> - For the qualitative comparison in Fig. 4, we respectfully suggest the reviewer recheck the supplementary **video** material. It shows not only an improvement in contrast. The previous methods tend to reconstruct misty images with distinct artifacts, especially in the region with insufficient event data. Our method effectively exploits the semantic prior of the diffusion model with the guidance of text prompts, which demonstrates semantic consistency. Additionally, Fig. 1 shows another extreme case. While the previous method (i.e., HyperE2VID) fails in the regions without event data, our method still reconstructs a reasonable scene according to the text guidance, which is closer to the ground truth. Tab. A-6 also shows the quantitative comparison in this dataset (i.e., HS-ERGB) and demonstrates the superiority of our method.
> - Regarding the artifacts in the letter “g”, as also noted by reviewer ***uWG8***, these issues arise because our method relies on the prior from the pre-trained diffusion model (SD2 [42]), which also struggles with text rendering. However, the recently released SD3 [79] claims to have improved text rendering capabilities, which could potentially address this problem and improve the performance.

---

### Official Review · Reviewer_fQWS · 2024-07-14

**Soundness:** 3
**Presentation:** 3
**Contribution:** 3
**Rating:** 4
**Confidence:** 5

**Summary:**

This paper uses abundant semantic information and raw event information to guide the reconstruction of RGB images from event images based on U-Net. Furthermore, this paper introduces event-aware mask loss to ensure temporal coherence and a noise initialization strategy to enhance spatial consistency. Experiments demonstrate that the proposed algorithm has a strong reconstruction performance.

**Strengths:**

1. Constructing ESA utilizes abundant semantic information and raw events to construct the cross attention with the combination of frames and raw events separately to guide event image reconstruction to RGB image.

2. The mask loss is constructed to supervise the reconstructed image from the temporal dimension strongly.

3. Extensive experiments on three datasets covering diverse challenging scenarios (e.g., fast motion, low light) demonstrate the superiority of this method.

**Weaknesses:**

1. In the ESA module, two modalities of information, raw events and text were used for cross-attention with frames. Still, no ablation experiments were given, which makes it impossible to determine whether the final experimental results of this work are more useful for raw events or text. In particular, raw events were inserted in each U-Net section.

2. In this paper, there are no reported flops as well as parameters, especially relative to some of the second-best methods in Table 1, what is the approximate percent increase in the two parameters?

**Questions:**

1. What are the tagging models mentioned in the paper? There are no specifics and no ablation experiments.

2. The comparison methods in the references are mostly from 2020 and 2021, with only one method from 2024.

3. Other methods are compared on SSIM metrics. Why not give SSIM metrics instead of SSIM* metrics?

4. How do you define spatial consistency? Adding ‘Noise Initialization’ only in the testing phase did not enhance the consistency of the model itself in the spatial dimension.

**Limitations:**

See Weaknesses. The final rating will be made based on the rebuttal.

---

> ### Author Rebuttal · Authors · 2024-08-07
>
> Thank the reviewer for valuable comments and suggestions. We address the concerns below:
>
> > **About ablation experiments for ESA on raw events and text.**
> >
> - The reviewer might misunderstand the ESA module. Indeed, the ESA and the text are **separate** parts in our framework. As outlined in Sec. 4.2 (Line 157-183), the ESA module consists of event spatial attention and event temporal attention layers, which compute attention based on the hidden state $z$ and event feature $z_e$, respectively. This way, ESA module facilitates spatio-temporal consistency between event data and video. The text input is integrated separately via a cross-attention layer that follows the original LDM [42], and it is not involved in the ESA module. Since event input is essential for the E2V reconstruction task, we have already provided ablation studies on both ESA and text with event input to demonstrate their individual effectiveness. Hereby, we reiterate the results as follows:
>     - To demonstrate the impact of text description, the ablation study in Tab. 2 (1st row vs. 3rd row) and Fig. 7 (left) compare the E2V results with (w) and without (w/o) text. Please check them.
>     - Tab. 3 (1st row) compares the performance of the baseline trained with simple channel-wise concatenation of events, without the ESA module,  showing a significant performance drop. This ablation result confirms the effectiveness of ESA module.
>
> > **About FLOPs and parameters comparison.**
> >
> - Thanks for the suggestion. Please refer to **Global Response to All Reviewers**.
>
> > **About the tagging models and the corresponding ablation experiments.**
> >
> - In Line 216, we employ the off-the-shelf tagging model RAM [61], which serves as a prompting model to provide text descriptions for the datasets. In fact, a recent work (SeeSR [70]) has also demonstrated the superiority of RAM compared to other models because of **rich objects and concise description**.
> - As suggested, we also tested BLIP[71] on a sampled sequence (i.e., *boxes* in HQF) to further evaluate the influence of the prompting model, as shown in Tab. A-2. BLIP can generate reasonable text prompts in caption-style and show **nearly reconstructed performance** on MSE (0.025 vs 0.027). A detailed discussion will be provided in the camera-ready revision.
>
> Table A-2: Comparisons between different prompting models on *boxes* of HQF.
>
> | Models | MSE | SSIM | LPIPS |
> | --- | --- | --- | --- |
> | RAM [61] | 0.025 | 0.557 | 0.196 |
> | BLIP [71] | 0.027 | 0.546 | 0.207 |
>
> > **Additional comparison methods.**
> >
> - We mainly compared with the latest state-of-the-art method, e.g., HyperE2VID (TIP 2024). By default, our method is superior to the methods published in 2022 and 2023. However, upon the suggestion, we have provided more comparison methods in Tab. A-3, which further demonstrates the superior performance of our method. We will update Tab.1 of main paper in the camera-ready revision.
>
> Table A-3: Comparison with more event-to-video methods.
>
> | Methods | | ECD | | | MVSEC | | | HQF | |
> | --- | --- | --- | --- | --- | --- | --- | --- | --- | --- |
> |  | MSE | SSIM | LPIPS | MSE | SSIM | LPIPS | MSE | SSIM | LPIPS |
> | Zhang et.al. (TPAMI2022) [72] | 0.076 | 0.519 | 0.457 | - | - | - | - | - | - |
> | EVSNN (CVPR2022) [73] | 0.061 | 0.570 | 0.362 | 0.104 | 0.389 | 0.538 | 0.086 | 0.482 | 0.433 |
> | PA-EVSNN (CVPR2022) [73] | 0.046 | 0.626 | 0.367 | 0.107 | **0.403** | 0.566 | 0.061 | 0.532 | 0.416 |
> | CISTA-LSTC (TPAMI2023) [74] | 0.038 | 0.585 | 0.229 | - | - | - | 0.041      | 0.563 | 0.271 |
> | CISTA-Flow (Arxiv2024) [75] | 0.047 | 0.586 | 0.225 | - | - | - | 0.034      | **0.590** | 0.257 |
> | HyperE2VID (TIP2024) [12] | 0.033 | 0.576 | 0.212 | 0.076 | 0.315 | 0.476 | **0.031** | 0.531 | 0.257 |
> | LaSe-E2V (OUrs) | **0.023** | **0.629** | **0.194** | **0.055** | 0.342 | **0.461** | 0.034 | 0.548 | **0.254** |
>
> > **About SSIM and SSIM\* metric.**
> >
> - The SSIM metric raises ambiguity because it involves several hyper-parameters that may differ across various codebases. For example, in the `structural_similarity` function of the **skimage** package, parameters like `gaussian_weights` and `sigma` are used for spatial weighting of each patch with a Gaussian kernel, `use_sample_covariance` indicates whether to normalize covariances, and `K1` and `K2` are algorithm-specific parameters that need to be set. For this reason, we **reevaluated** all comparison methods by using a unified metric, denoted as the SSIM* scores. We will further clarify this point in the revision.
>
> > **How to define spatial consistency and how does noise initialization enhance the consistency?**
> >
> - Spatial consistency denotes the consistency between event data and the reconstructed video. Specifically, events typically occur at the **edges/texture** part of the scene context, and the reconstructed video needs to align the event data in the spatial structure.
> - Noise Initialization mainly focuses on mitigating the **train-test gap** during inference, which is a common challenge in diffusion models. Intuitively, accumulated event data provides structural information (e.g. edges) in the scene, acting as an **additional constraint** during the denoising process.
>
> **Additional Reference:**
>
> [71] Li J, Li D, Savarese S, et al. Blip-2: Bootstrapping language-image pre-training with frozen image encoders and large language models[C]//ICML, 2023.
>
> [72] Zhang Z, Yezzi A J, Gallego G. Formulating event-based image reconstruction as a linear inverse problem with deep regularization using optical flow[J]. TPAMI, 2022, 45(7): 8372-8389.
>
> [73] Zhu L, Wang X, Chang Y, et al. Event-based video reconstruction via potential-assisted spiking neural network[C]//CVPR. 2022.
>
> [74] Liu S, Dragotti P L. Sensing diversity and sparsity models for event generation and video reconstruction from events[J]. TPAMI, 2023, 45(10): 12444-12458.
>
> [75] Liu S, Dragotti P L. Enhanced Event-Based Video Reconstruction with Motion Compensation[J]. arXiv, 2024.

---

### Author Rebuttal · Authors · 2024-08-07

### **Global Response to All Reviewers**

We sincerely thank the reviewers for their constructive feedback. We are pleased that the reviewers found our **method to be novel and effective**, the **performance to be strong and with high-quality**, which is believed to **inspire future work** in the community. Below, we address the questions raised and promise to throughly revise the paper accordingly.

> **Complexity analysis**
>
- Following the reviewers' suggestions, we have provided a detailed computational complexity analysis in Tab. A-1, which includes recent event-to-video reconstruction methods and related diffusion-based approaches. Our method requires a significant amount of inference time due to the 50 denoising steps, in contrast to previous single-step event-to-video methods. However, we kindly note that this is a common challenge for all diffusion-based models, as observed in diffusion-based super-resolution [69,70] and depth estimation methods [66,67]. To reduce inference time, some research, eg., [65], already explored to decrease the number of denoising steps, offering a promising direction for further improvement (as a future work) to our framework.
- Please be noted that previous approaches [54, 12]  often struggle to recover regions *without active events*. In contrast, as demonstrated in Fig.1, our method achieves **holistic, semantic-aware reconstruction**. As acknowledged by reviewers ***ueNs*** and ***uWG8***, this paper aims to *"**provide new ideas for subsequent works**"* and *"**inspire future work in the field**"* by incorporating language guidance.
- As affirmed by reviewer ***H8fU***, this paper demonstrates that "***it is possible to use diffusion-based models for the event-based video reconstruction task***," thereby leveraging the rich semantic priors of large-scale LDM. This approach holds promise for extending to other event-based tasks, including video frame interpolation, deblurring, and denoising.

In summary, albeit with lower inference speed than conventional E2V methods (but higher than diffusion-based super-resolution and depth estimation methods), our work brings new ideas and may hopefully inspire new future research for event-based vision.

Table A-1: Complexity comparison on various methods. All tests are conducted on one NVIDIA Tesla 32G-V100 GPU.

| Methods |  | Parameters | Inference time (per frame) |
| --- | --- | --- | --- |
| Conventional Event-to-Video | ET-Net [54] | 22.18M | 0.0124s |
|  | HyperE2VID [12] | 10.15M | 0.0043s |
| Diffusion-based Depth Estimation | DepthFM [66] | 891M | 2.1s |
|  | Marigold [67] | 948M | 5.2s |
| Diffusion-based Super-Resolution | StableSR [68] | 1409M | 18.70s |
|  | PASD [69] | 1900M | 6.07s |
|  | SeeSR [70] | 2284M | 7.24s |
| Diffusion-based Event-to-Video | **LaSe-E2V (Ours)** | 1801M | 1.09s |

**Additional Reference**:

[65] Liu X, Zhang X, Ma J, et al. Instaflow: One step is enough for high-quality diffusion-based text-to-image generation[C]//ICLR. 2023.

[66] Gui M, Fischer J S, Prestel U, et al. Depthfm: Fast monocular depth estimation with flow matching[J]. arXiv, 2024.

[67] Ke B, Obukhov A, Huang S, et al. Repurposing diffusion-based image generators for monocular depth estimation[C]//CVPR. 2024.

[68] Wang J, Yue Z, Zhou S, et al. Exploiting diffusion prior for real-world image super-resolution[J]. IJCV, 2024: 1-21.

[69] Yang T, Ren P, Xie X, et al. Pixel-aware stable diffusion for realistic image super-resolution and personalized stylization[J]. arXiv, 2023.

[70] Wu R, Yang T, Sun L, et al. Seesr: Towards semantics-aware real-world image super-resolution[C]//CVPR. 2024.

---

### Comment · Area_Chair_SGLG · 2024-08-10

Hi reviewers,

Thank you for your hard work in reviewing the paper! Please check out the authors' responses and ask any questions you have to help clarify things by Aug 13.

--AC

---

### Decision · Program_Chairs · 2024-09-25

**Decision:**

Accept (poster)

**Comment:**

This paper is at borderline with ratings of 1x borderline reject, 1x borderline accept and 2x weak accepts. The reviewers mention that the strengths of this paper is that it is the first to tackle tackle the event data reconstruction task from a language-guided perspective. This approach could inspire future work in the field. The proposed algorithm achieves optimal performance on nearly all metrics across multiple datasets, demonstrating superior visual effects and validating the algorithm's effectiveness. The paper presents a high-quality body of work, including effective methods and extensive experimental validation. This paper proves that it is possible to use diffusion-based models for the event-based video reconstruction task. The event-guided spatio / temporal attention provides a new idea for event-image fusion. The reviewer who gave weak reject mentioned that no ablation given for the ESA module and no reported flops and parameters are given in Tab. 1 of the paper. These concerns are addressed in the authors' rebuttal. On the consideration of the strengths outweighing the weaknesses, the metareviewers decide to accept the paper.